# Episodic Multi-agent Reinforcement Learning with Curiosity-driven Exploration

**Lulu Zheng**[*1], **Jiarui Chen**[*2 3], **Jianhao Wang**[1], **Jiamin He**[4†], **Yujing Hu**[3], **Yingfeng Chen**[3], **Changjie Fan**[3], **Yang Gao**[2], **Chongjie Zhang**[1]

[1]Institute for Interdisciplinary Information Sciences, Tsinghua University, China
[2]Department of Computer Science and Technology, Nanjing University, China
[3]Fuxi AI Lab, NetEase, China
[4]Department of Computing Science, University of Alberta, Canada

zll19@mails.tsinghua.edu.cn
chenjiarui@smail.nju.edu.cn
wjh19@mails.tsinghua.edu.cn
jiamin12@ualberta.ca
{huyujing, chenyingfeng01, fanchangjie}@corp.netease.com
gaoy@nju.edu.cn
chongjie@tsinghua.edu.cn

## Abstract

Efficient exploration in deep cooperative *multi-agent reinforcement learning* (MARL) still remains challenging in complex coordination problems. In this paper, we introduce a novel Episodic Multi-agent reinforcement learning with Curiosity-driven exploration, called EMC. We leverage an insight of popular factorized MARL algorithms that the "induced" individual Q-values, i.e., the individual utility functions used for local execution, are the embeddings of local action-observation histories, and can capture the interaction between agents due to reward backpropagation during centralized training. Therefore, we use prediction errors of individual Q-values as intrinsic rewards for coordinated exploration and utilize episodic memory to exploit explored informative experience to boost policy training. As the dynamics of an agent's individual Q-value function captures the novelty of states and the influence from other agents, our intrinsic reward can induce coordinated exploration to new or promising states. We illustrate the advantages of our method by didactic examples, and demonstrate its significant outperformance over state-of-the-art MARL baselines on challenging tasks in the StarCraft II micromanagement benchmark.

## 1 Introduction

Cooperative *multi-agent reinforcement learning* (MARL) has great promise to solve many real-world multi-agent problems, such as autonomous cars [1] and robots [2]. These complex applications post two major challenges for cooperative MARL: *scalability*, i.e., the joint-action space exponentially grows as the number of agents increases, and *partial observability*, which requires agents to make decentralized decisions based on their local action-observation histories due to communication constraints. Luckily, a popular MARL paradigm, called *centralized training with decentralized execution* (CTDE) [3], is adopted to deal with these challenges. With this paradigm, agents' policies are trained with access to global information in a centralized way and executed only based on local

---

[*]Equal contribution.
[†]Work performed while visiting Tsinghua Univeristy.

35th Conference on Neural Information Processing Systems (NeurIPS 2021).

histories in a decentralized way. Based on the paradigm of CTDE, many deep MARL methods have been proposed, including VDN [4], QMIX [5], QTRAN [6], and QPLEX [7].

A core idea of these approaches is to use value factorization, which uses neural networks to represent the joint state-action value as a function of individual utility functions, which can be referred to *individial Q-values* for terminological simplicity. For example, VDN learns a centralized but factorizable joint value function $Q_{tot}$ represented as the summation of individual value function $Q_i$. During execution, the decentralized policies can be easily derived for each agent $i$ by greedily selecting actions with respect to its local value function $Q_i$. By utilizing this factorization structure, an implicit multi-agent credit assignment is realized because $Q_i$ is represented as a latent embedding and is learned by neural network backpropagation from the total temporal-difference error on the single global reward signal, rather than on a local reward signal specific to agent $i$. This value factorization technique enables value-based MARL approaches, such as QMIX and QPLEX, to achieve state-of-the-art performance in challenging tasks such as the StarCraft unit micromanagement [8].

Despite the current success, since only using simple $\epsilon$-greedy exploration strategy, these deep MARL approaches are found ineffective to solve complex coordination tasks that require coordinated and efficient exploration [7]. Exploration has been extensively studied in single-agent reinforcement learning and many advanced methods have been proposed, including pseudo-counts [9, 10], curiosity [11, 12], and information gain [13]. However, these methods cannot be adopted into MARL directly, due to the exponentially growing state space and partial observability, leaving multi-agent exploration challenging. Recently, only a few works have tried to address this problem. For instance, EDTI [14] uses influence-based methods to quantify the value of agents' interactions and coordinate exploration towards high-value interactions. This approach empirically shows promising results but, because of the need to explicitly estimate the influence among agents, it is not scalable when the number of agents increases. Another method, called MAVEN [15], introduces a hierarchical control method with a shared latent variable encouraging committed, temporally extended exploration. However, since the latent variable still needs to explore in the space of joint behaviours [15], it is not efficient in complex tasks with large state spaces.

In this paper, we propose a novel multi-agent curiosity-driven exploration method. Curiosity is a type of intrinsic motivation for exploration, which usually uses prediction errors on different spaces (e.g., future observations [12], actions [11], or learnable representation [16]) as a reward signal. Recently, curiosity-driven methods have achieved significant success in single-agent reinforcement learning [12, 17, 18]. However, curiosity-driven methods face a critical challenge in MARL: in which space should we define curiosity? The straightforward method is to measure curiosity on the global observation [12] or joint histories in a centralized way. However, it is inefficient to find structured interaction between agents, which seems too sparse compared with the exponentially growing state space when the number of agents increases. In contrast, if curiosity is defined as the novelty of local observation histories during the decentralized execution, although scalable, it still fails to guide agents to coordinate due to partial observability. Therefore, we find a middle point of centralized curiosity and decentralized curiosity, i.e., utilizing the value factorization of the state-of-the-art multi-agent Q-learning approaches and defining the prediction errors of individual Q-value functions as intrinsic rewards.

The significance of this intrinsic reward is two-fold: 1) it provides a novelty measure of joint observation histories with scalability because individual Q-values are latent embeddings (i.e., an effective state abstraction [19]) of observation histories in factorized multi-agent Q-learning (e.g., VDN or QPLEX); and 2) as shown in Figure 1, it captures the influence from other agents due to the implicit credit assignment from global reward signal during centralized training [20], and biases exploration into promising

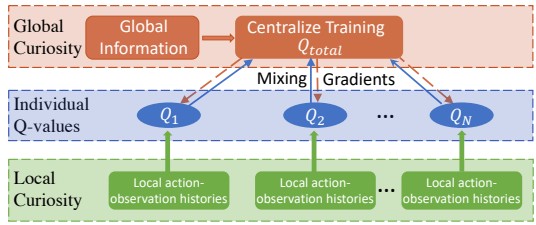

Figure 1: CTDE Framework

states where strong interdependence may lie between agents. Therefore, with this novel intrinsic reward, our curiosity-driven method enables efficient, diverse, and coordinated exploration for deep multi-agent Q-learning with value factorization.

Besides efficient exploration, another challenge for deep MARL approaches is how to make the best use of experiences collected by the exploration strategy. Prioritized experience replay based on TD errors shows effectiveness in single-agent deep reinforcement learning. However, it does

not carry this promise in factorized multi-agent Q-learning, since the projection error induced by value factorization is also fused into the TD error and severally degrades the effectiveness of the TD error as a measure of the usefulness of experiences. To efficiently use promising exploratory experience trajectories, we augment factorized multi-agent reinforcement learning with episodic memory [21, 22]. This memory stores and regularly updates the best returns for explored states. We use the results in the episodic memory to regularize the TD loss, which allows fast latching onto past successful experience trajectories collected by curiosity-driven exploration and greatly improves learning efficiency. Therefore, we call our method Episodic Multi-agent reinforcement learning with Curiosity-driven exploration, called EMC.

We evaluate EMC in didactic examples, and a broad set of StarCraft II micromanagement benchmark tasks [8]. The didactic examples along with detailed visualization illustrate that our proposed intrinsic reward can guide agents' policies to novel or promising states, thus enabling effectively coordinated exploration. Empirical results on more complicated StarCraft II tasks show that EMC significantly outperforms other multi-agent state-of-the-art baselines.

## 2 Background

### 2.1 Dec-POMDP

A cooperative multi-agent task can be modelled as a Dec-POMDP [23], which is defined by a tuple $G =< \mathcal{I}, \mathcal{S}, \mathcal{A}, P, R, \Omega, O, n, \gamma >$, where $\mathcal{I}$ is the sets of $n$ agents, $\mathcal{S}$ is the global state space, $\mathcal{A}$ is the finite action set, $\gamma \in [0, 1)$ is the discount factor. We consider a partially observable setting in a Dec-POMDP, i.e., at each timestep, agent $i \in \mathcal{I}$ only has access to the observation $o_i \in \Omega$ drawn from the observation function $O(s, i)$. Besides, each agent has an action-observation history $\tau_i \in \mathcal{T} \equiv (\Omega \times \mathcal{A})^* \times \Omega$ and constructs its individual policy to jointly maximize team performance. With each agent $i$ selecting an action $a_i \in \mathcal{A}$, the joint action $\boldsymbol{a} \equiv [a_i]_{i=1}^n \in \boldsymbol{\mathcal{A}} \equiv \mathcal{A}^N$ leads to a shared reward $r = R(s, \boldsymbol{a})$ and the next state $s'$ according to the transition distribution $P(s'|s, \boldsymbol{a})$. The formal objective function is to find a joint policy $\boldsymbol{\pi}$ that maximizes a joint value function $V^{\boldsymbol{\pi}}(s) = \mathbb{E}[\sum_{t=0}^\infty \gamma^t r_t | s = s_0, \boldsymbol{\pi}]$, or a joint action-value function $Q^{\pi}(s, \boldsymbol{a}) = r(s, \boldsymbol{a}) + \gamma \mathbb{E}_{s'}[V^{\boldsymbol{\pi}}(s')]$.

### 2.2 Centralized Training With Decentralized Execution (CTDE)

CTDE is a promising paradigm in deep cooperative multi-agent reinforcement learning [3, 23, 24], where the local agents execute actions only based on local observation histories, while the policies can be trained in centralized manager which has access to global information. During the training process, the whole team cooperate to find the optimal joint action-value function $Q_{tot}^*(s, \boldsymbol{a}) = r(s, \boldsymbol{a}) + \gamma \mathbb{E}_{s'}[\max_{\boldsymbol{a}'} Q_{tot}^*(s', \boldsymbol{a}')]$. Due to partial observability, we use $Q_{tot}(\boldsymbol{\tau}, \boldsymbol{a}; \boldsymbol{\theta})$ instead of $Q_{tot}(s, \boldsymbol{a}; \boldsymbol{\theta})$, where $\boldsymbol{\tau} \in \boldsymbol{\mathcal{T}} \equiv \mathcal{T}^N$. Then the Q-value neural network will be trained to minimize the following expected TD-error:

$$\mathcal{L}(\boldsymbol{\theta}) = \mathbb{E}_{\boldsymbol{\tau}, \boldsymbol{a}, \boldsymbol{r}, \boldsymbol{\tau}' \in D} \left[ r + \gamma V(\boldsymbol{\tau}'; \boldsymbol{\theta}^-) - Q_{tot}(\boldsymbol{\tau}, \boldsymbol{a}; \boldsymbol{\theta}) \right]^2, \tag{1}$$

where $D$ is the replay buffer and $\boldsymbol{\theta}^-$ denotes the parameters of the target network, which is periodically updated by $\boldsymbol{\theta}$. And $V(\boldsymbol{\tau}'; \boldsymbol{\theta}^-)$ is the one-step expected future return of the TD target. Local agents can only obtain local action-observation history and need inference based on individual Q-value functions $Q_i(\tau_i, a_i)$. Therefore, many works have made efforts in finding the factorization structures between joint Q-value functions $Q_{tot}$ and individual Q-functions $Q_i(\tau_i, a_i)$ [4, 5, 7].

## 3 Related Work

**Curiosity-driven Exploration** Curiosity-driven exploration has been well studied in single-agent reinforcement learning. Previous literature [25, 26] has provided a good summary in this topic. Recently, curiosity-driven methods have made great progress in deep reinforcement learning. For example, some works use pseudo-state counts to get intrinsic rewards [9, 10, 27] instead of count-based methods to get better scalability. Stadie et al. [28] use prediction errors in the feature space of an auto-encoder to measure the novelty of states and encourage exploration. On the other hand, Mohamed and Rezende [29] propose to use empowerment, measured by the information gain based on the entropy of actions, as intrinsic rewards for exploring novel states efficiently. Another information-based method [13] tries to maximize information gain about the agent's belief of the environment's

dynamics as an exploration strategy. ICM [11] learns an inverse model which predicts the agent's action given its current and next states and tries to predict the next state in the learned hidden space by current state and action. RND [12] uses curiosity as intrinsic rewards in a simpler but effective way, which uses a fixed randomly initialized neural network as a representation network and directly predicts the embedding of the next state. Different from these methods, we are the first to propose an advanced curiosity-driven exploration method in MARL setting for diverse and coordinated exploration.

**Multi-agent Exploration** Although single-agent exploration is extensively studied and has achieved considerable success, few exploration methods were designed for cooperative MARL. Bargiacchiet al. [30] proposes an exploration method that can only be used in repeated single-stage problems. Jaques et al. [31] defines intrinsic reward by "social influence" to encourage agents to choose actions that can influence other agents' actions. Iqbal and Sha [32] uses various simple exploration methods to learn simultaneously and then put the samples of every method in a shared buffer to achieve the coordinated exploration. Wang et al. [14] use mutual information (MI) to capture the interdependence of the rewards and transitions between agents. MAVEN [15] is the state-of-the-art exploration method in MARL that uses a hierarchical policy to produce a shared latent variable and learns several state-action value functions for each agent. These works, although important, still face the challenge of achieving scalable and effective multi-agent exploration.

**Episodic Control** Our work is also related to episodic control reinforcement learning, which is usually adopted in single-agent settings for better sample efficiency. Previous works propose to use episodic memory in near-deterministic environment [33–36]. Model-free episodic control [34] uses a completely non-parametric table to keep the best Q-values of state-action pair in a tabular-based memory and uses a k-nearest-neighbors fashion to find the sequence of actions that so far yielded the highest return from a given start state in the memory. Recently, several extensions have been proposed to integrate episodic control with parametric DQN. Gershman and Daw [37] uses episodic memory to retrieve samples and then average future returns to approximate the action values. EMDQN [21] uses a fixed random matrix as a representation function and uses the projection of states as keys to store the information of episodic memory into a non-parametric model. Using the episodic-memory based target as a regularization term to guide the training process, the performance of EMDQN is significantly improved compared with the original DQN. Despite the fruitful progress made in single-agent episodic reinforcement learning, few works study episodic control in a multi-agent setting. To the best of our knowledge, we are the first to utilize the mechanism of episodic control in deep multi-agent reinforcement learning.

## 4 Episodic Multi-agent Reinforcement Learning with Curiosity-Driven Exploration

In this section, we introduce EMC, a novel episodic multi-agent exploration framework. EMC takes prediction errors of individual Q-value functions as intrinsic rewards for guiding the diverse and coordinated exploration. After collecting informative experience, we leverage an episodic memory to memorize the highly rewarding sequences and use it as the reference of a one-step TD target to boost multi-agent Q-learning. First, we analyze the motivations for predicting individual Q-values, then we introduce the curiosity module for exploration. Finally, we describe how to utilize episodic memory to boost training.

### 4.1 Curiosity-Driven Exploration by Predicting Individual Q-values

As shown in Figure 2, in the paradigm of CDTE, local agents make decisions based on individual Q-value functions, which take local observation histories as inputs, and are updated by the centralized module which has access to global information for training. The key insight is that, different from single-agent cases, individual Q-value functions in MARL are used for both decision-making and embedding historical observations. Furthermore, due to implicit credit assignment by global reward signal during centralized training, individual Q-value functions $Q_i(\tau_i, \cdot)$ are influenced by environment as well as other agents' behaviors. More concretely, it has been proved by Wang et al. [20] that, when the joint Q-function $Q_{tot}$ is factorized into linear combination of individual Q-functions $Q_i$, i.e., $Q_{tot}^{(t+1)}(\boldsymbol{\tau}, \boldsymbol{a}) = \sum_{i=1}^{N} Q_i^{(t+1)}(\tau_i, a_i)$, then $Q_i^{(t+1)}(\tau_i, a_i)$ has the following closed-form solution:

$$Q_i^{(t+1)}(\tau_i, a_i) = \underbrace{\mathbb{E}_{(\tau'_{-i}, a'_{-i}) \sim p_D(\cdot|\tau_i)} \left[ y^{(t)} \left( \tau_i \oplus \tau'_{-i}, a_i \oplus a'_{-i} \right) \right]}_{\text{evaluation of the individual action } a_i}$$

$$-\frac{n-1}{n} \underbrace{\mathbb{E}_{\tau', a' \sim p_D(\cdot|\Lambda^{-1}(\tau_i))} \left[ y^{(t)} \left( \tau', a' \right) \right]}_{\text{counterfactual baseline}} + w_i(\tau_i), \qquad (2)$$

where $y^{(t)}(\tau, a) = r + \gamma \mathbb{E}_{\tau'} \left[ \max_{a'} Q_{tot}^{(t)}(\tau', a') \right]$ denotes the expected one-step TD target, and $p_D(\cdot|\tau_i)$ denotes the conditional empirical probability of $\tau_i$ in the given dataset $D$. The notation $\tau_i \oplus \tau'_{-i}$ denotes $\langle \tau'_1, \dots, \tau'_{i-1}, \tau_i, \tau'_{i+1}, \dots, \tau'_n \rangle$, and $\tau'_{-i}$ denotes the elements of all agents except for agent $i$. $\Lambda^{-1}(\tau_i)$ denotes the set of trajectory histories that may share the same latent-state trajectory as $\tau_i$. The residue term $\boldsymbol{w} \equiv [w_i]_{i=1}^n$ is an arbitrary function satisfying $\forall \tau \in \Gamma, \sum_{i=1}^n w_i(\tau_i) = 0$.

Eq. (2) shows that by *linear value factorization*, the individual Q-value $Q_i(\tau_i, a_i)$ is not only decided by local observation histories but also influenced by other agents' action-observation histories. Thus predicting $Q_i$ can capture both the novelty of states and the interaction between agents and lead agents to explore promising states. Motivated by this insight, in this paper, we use a linear value factorization module separate from the inference module to learn the individual value function $Q_i$, and use the prediction errors of $Q_i$ as intrinsic rewards to guide exploration. In this paper, we define the prediction errors of individual Q-values as *curiosity* and propose our curiosity-driven exploration module as follows.

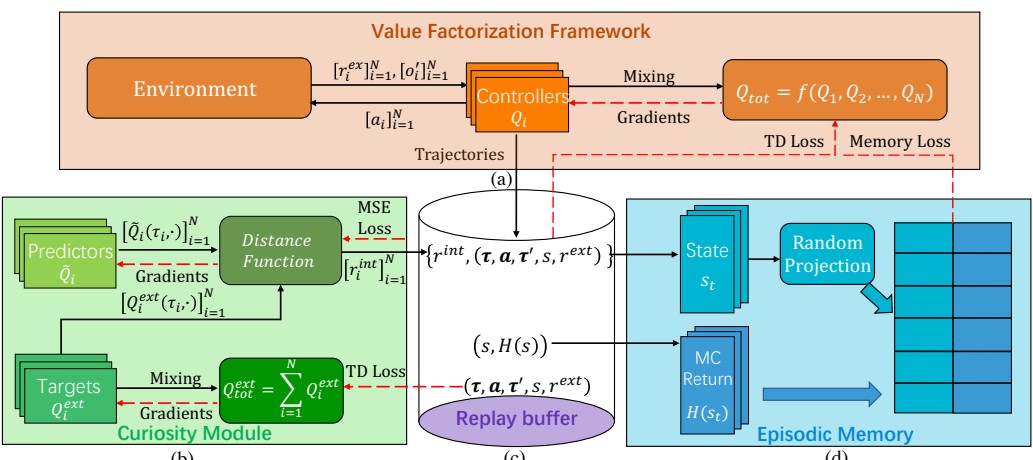

Figure 2: An overview of EMC's framework

Figure 2b demonstrates the *Curiosity Module*, separated from the inference module (Figure 2a). The curiosity module consists of four components: (i) The centralized training part with linear value factorization, which shares the same implementation as VDN [4], but only trained with extrinsic rewards $r^{ext}$ from the environment; (ii) the *Target* for prediction, i.e., the corresponding individual Q-values $Q_i^{ext}$, represented by a recurrent Q-network; (iii) *Predictor* $\widetilde{Q}_i(\tau_i)$, which is used for predicting $Q_i^{ext}$ and shares the same network architecture as *Target* $Q_i^{ext}$; and (iv) *Distance Function*, which measures the distance between $Q_i^{ext}$ and $\widetilde{Q}_i$, e.g., $L_2$ distance. The predictors are trained by minimizing the Mean Squared Error (MSE) of the distance in an end-to-end manner. For stable training, we use the soft-update target [38] of $Q_i^{ext}$ to smooth the outputs of the targets. In general, (ii) is trained with (i) and outputs individual Q-values , while (iii) is trained with (ii) and (iv), and aims to predict the soft-update target of individual Q-values. Motivated by the implicit credit assignment of linear value factorization (Eq. (2)), the curiosity module predicts the individual Q-values $[Q_i^{ext}]_{i=1}^n$ in linear factorization, i.e., $Q_{tot}^{ext} = \sum_{i=1}^N Q_i^{ext}$. Then the curiosity-driven intrinsic reward is generated by the following equation:

$$r^{int} = \frac{1}{N} \sum_{i=1}^N \left\| \widetilde{Q}_i(\tau_i, \cdot) - Q_i^{ext}(\tau_i, \cdot) \right\|_2, \qquad (3)$$

This intrinsic reward is used for the centralized training of the inference module, as shown in Figure 2a:

$$\mathcal{L}_{\text{inference}}(\boldsymbol{\theta}) = \mathbb{E}_{\boldsymbol{\tau},\boldsymbol{a},\boldsymbol{r},\boldsymbol{\tau}' \in D} \left[ (y(\boldsymbol{\tau},\boldsymbol{a}) - Q_{tot}(\boldsymbol{\tau},\boldsymbol{a};\boldsymbol{\theta}))^2 \right], \tag{4}$$

where $y(\boldsymbol{\tau},\boldsymbol{a}) = r^{ext} + \beta r^{int} + \gamma \max_{\boldsymbol{a}'} Q_{tot}(\boldsymbol{\tau}',\boldsymbol{a}';\boldsymbol{\theta}^-))$, denoting one step TD target of the inference module, and $\beta$ is the weight term of the intrinsic reward. We use a separate training model for inference (Figure 2a) to avoid the accumulation of projection errors of $Q_i$ during training.

The independence of inference module leads to another advantage, that EMC's architecture can be adopted into many value-factorization-based multi-agent algorithms which utilize the CDTE paradigm, i.e., the general function $f$ in Figure 2a can indicate specific (linear, monotonic and IGM) value factorization structures in VDN [4], QMIX [5], and QPLEX [7], respectively. In this paper, we utilize these state-of-the-art algorithms for the inference module. With this curiosity-driven bias plugged into ordinary MARL algorithms, EMC will achieve efficient, diverse and coordinated exploration.

## 4.2 Episodic Memory

Equipped with efficient exploration ability, another challenge is how to make the best use of good trajectories collected by exploration effectively. Recently, episodic control has been widely studied in single-agent reinforcement learning [21, 22], which can replay the highly rewarding sequences, thus boosting training. Inspired by this framework, we generalize single-agent episodic control to propose a multi-agent episodic memory, which records the best memorized Monte-Carlo return in the episode, and provide a memory target $H$ as a reference to regularize the ordinary one-step inference TD target estimation in the inference module (Figure 2a):

$$\mathcal{L}_{\text{memory}}(\boldsymbol{\theta}) = \mathbb{E}_{\boldsymbol{\tau},\boldsymbol{a},\boldsymbol{r},\boldsymbol{\tau}' \in D} \left[ (H - Q_{tot}(\boldsymbol{\tau},\boldsymbol{a};\boldsymbol{\theta}))^2 \right]. \tag{5}$$

However, different from the single-agent episodic control, the action space of MARL exponentially grows as the number of agents increases, and partial observability also limits the information of individual value functions. Thus, we maintain our episodic memory by storing the state-value function on the global state space and utilizing the global information during the centralized training process under the CTDE paradigm. Figure 2d shows the architecture of the *Episodic Memory*. We keep a memory table $M$ to record the maximum remembered return of the current state, and use a fixed random matrix drawn from Gaussian distribution as a representation function to project states into low-dimensional vectors $\phi(s) : S \to \mathbb{R}^k$, which are used as keys to look up corresponding global state value function $H(\phi(s_t))$. When our exploration method collects a new trajectory, we update our memory table $M$ as follows:

$$H(\phi(s_t)) = \begin{cases} \max\{H(\phi(\hat{s}_t)), R_t(s_t, \boldsymbol{a}_t)\} & if \; \|\phi(\hat{s}_t) - \phi(s_t)\|_2 < \delta \\ R_t(s_t, \boldsymbol{a}_t) & otherwise \end{cases}, \tag{6}$$

where $\phi(\hat{s}_t)$ is $\phi(s_t)$'s nearest neighbor in the memory $M$, $\delta$ is a threshold, and $R(s_t, \boldsymbol{a}_t)$ represents the future return when agents taking joint action $\boldsymbol{a}_t$ under global state $s_t$ at the $t$-th timestep in a new episode. In our implementation, $\phi(s_t) \in M$ is indeed evaluated approximately based on the embedding distance. Specifically, when the key of the state $\phi(s_t)$ is close enough to one key in the memory, we assume that $\phi(s_t) \in M$ and find the best memorized Monte-Carlo return correspondingly. Otherwise, we think $\phi(s_t) \notin M$ and record the state's return into the memory. Leveraging the episodic memory, we can directly obtain the maximum remembered return of the current state, and use the one-step TD memory target $H$ as a reference to regularize learning:

$$H(\phi(s_t), \boldsymbol{a}_t) = r_t(s_t, \boldsymbol{a}_t) + \gamma H(\phi(s_{t+1})). \tag{7}$$

Thus, the new objective function for the inference module is:

$$\begin{aligned} \mathcal{L}_{\text{total}}(\boldsymbol{\theta}) &= \mathcal{L}_{\text{inference}}(\boldsymbol{\theta}) + \lambda \mathcal{L}_{\text{memory}}(\boldsymbol{\theta}) \\ &= \mathbb{E}_{\boldsymbol{\tau},\boldsymbol{a},\boldsymbol{r},\boldsymbol{\tau}' \in D} \left[ (y(\boldsymbol{\tau},\boldsymbol{a}) - Q_{tot}(\boldsymbol{\tau},\boldsymbol{a};\boldsymbol{\theta}))^2 + \lambda \left( H(\phi(s_t), \boldsymbol{a}_t) - Q_{tot}(\boldsymbol{\tau},\boldsymbol{a};\boldsymbol{\theta}))^2 \right], \end{aligned} \tag{8}$$

where $\lambda$ is the weighting term to balance the effect of episodic memory's reference. Using the maximum return from the episodic memory to propagate rewards, we can compensate for the disadvantage of slow learning induced by the original one-step reward update and improve sample efficiency.

# 5 Experiments

In this section, we will conduct a large set of empirical experiments for answering the following questions: (1) Is exploration by predicting individual Q-value functions better than exploration by decentralized curiosity or global curiosity (Section 5.1)? (2) Can our method perform efficient coordinated exploration in challenging multi-agent tasks (Section 5.2-5.3)? (3) If so, what role does each key component play for the superior performance (Section 5.4)? (4) Why do we choose to predict target $Q_i^{ext}$ for generating intrinsic rewards rather than other choices (Section 5.4)? We will propose several didactic examples and demonstrate the advantage of our method in coordinated exploration, and evaluate our method on the StarCraft II micromanagement (SMAC) benchmark [8] compared with existing state-of-the-art multi-agent reinforcement learning (MARL) algorithms: QPLEX [7], Weighted-QMIX [39], QTRAN [6], QMIX [5], VDN [4], RODE [40], and MAVEN [15].

## 5.1 Didactic Example

Figure 3 shows an $11 \times 12$ grid world game that requires coordinated exploration. The blue agent and the red agent can choose one of the five actions: *[up, down, left, right, stay]* at each time step. The wall shown in the picture isolates the two agents, and one agent cannot be observed by the other until it gets into the shaded area. The two agents will receive a positive global reward $r = 10$ if and only if they arrive at the corresponding goal grid (referred to the character $G$ in Figure 3) at the same time. If only one arrives, the incoordination will be punished by a negative reward $-p$.

To evaluate the effectiveness of our curiosity-driven exploration, we implement our method into QPLEX, QMIX, and VDN (denoted as *EMC-QPLEX*, *EMC-QMIX*, and *EMC-VDN*, respectively) and test them in this toy game compared with the state-of-the-art MARL algorithms: VDN, IQL, QMIX, and QPLEX. Moreover, to demonstrate the motivation of predicting individual Q-functions, we add two more baselines: QPLEX with the prediction error of global state as intrinsic rewards (denoted as *QPLEX-Global*), and QPLEX with the prediction error of local joint histories as intrinsic rewards (denoted as *QPLEX-Local*). Both of them use a fixed network to project the inputs into latent embedding, then predict the latent embedding to generate intrinsic reward, just like the

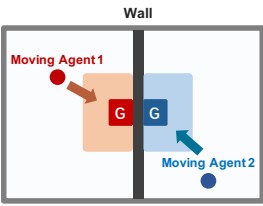

Figure 3: Coordinated Toygame

Random Network Distillation (RND) [12]. We test different punishment degrees, i.e., different $ps$ (which are deferred to Appendix C), and the results show QPLEX-Global and QPLEX-Local are effective enough for exploration when $p$ is relatively small. However, as $p$ increases, the task becomes more challenging since it requires sufficient and coordinated exploration. In Figure 4, we show the median test win rate of all methods over 6 random seeds when $p = 2$, and only our methods can learn the optimal policy and win the game, while other methods failed.

To understand this result better, we have made several visualisations to demonstrate our advantage in coordinated exploration. Figure 4 shows the heatmaps of visitation and intrinsic reward by EMC-QPLEX, QPLEX-Global, and QPLEX-Local. During the early stage of training, all methods uniformly explore the whole area (Figure 4a). As the exploration progresses, the global curiosity (QPLEX-Global) encourages agents to visit all configurations without bias, which is inefficient and fail to leverage the potential locality influence between agents (Figure 4b), resulting in extrinsic rewards beginning to dominate the behaviors (Figure 4c). On the other hand, the visitation heatmap of QPLEX-Local shows the decentralized curiosity encourages agents to explore around the goal grid, but it cannot promise to encourage agents to coordinate and gain the reward due to the partial observability in decentralized execution. In contrast, the heatmap of intrinsic reward for EMC-QPLEX shows that predicting individual Q-values will bias exploration into areas where individual Q-values are more dynamic due to the potential correlation between agents. Therefore, QPLEX-Local and QPLEX-Global both fail in this task (Figure 4c), while our methods are able to find the optimal policy. This didactic example shows the global curiosity or local curiosity may fail to handle complex tasks where coordinated exploration needs to be addressed. While since individual Q-values $Q_i$ are the embeddings of historical observations, and are dynamically updated by the backpropagation of the global reward signal gained through cooperation during centralized training. Thus $Q_i$ can implicitly reflect the influence from the environment and other agents, and predicting $Q_i$ can capture valuable and spare interactions among agents and bias exploration into new or promising states.

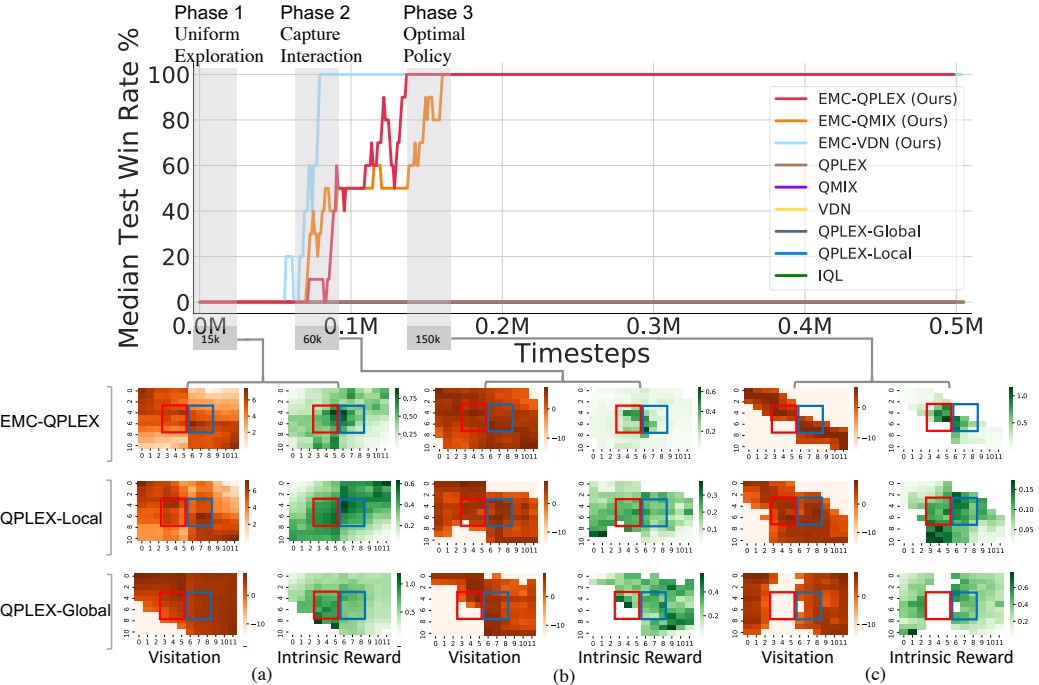

Figure 4: The heat map of gridworld game.

## 5.2 Predator Prey

Predator-Prey is a partially-observable multi-agent co-ordinated game with miscoordination penalties used by WQMIX [39]. As shown in Figure 5, since extensive exploration is needed to jump out of the local optima, WQMIX is the only baseline algorithm to find the optimal policy, due to its shaped data distribution which can be seen as a type of exploration. Other state-of-the-art multi-agent Q-learning algorithms, such as QPLEX and QMIX, fail to solve this task. For MAVEN, QPLEC-Global and QPLEX-Local, although equipped with improved exploration ability, they still failed to address coordination due to uniform exploration nature or partial observability. However, plugged with EMC, EMC-VDN, EMC-QMIX, and EMC-QPLEX can guarantee coordinated exploration effectively and achieve good performance.

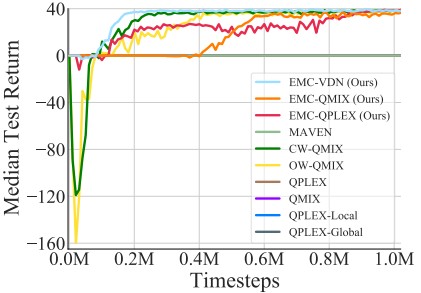

Figure 5: The performance of Predator Prey.

## 5.3 StarCraftII Micromanagement (SMAC) Benchmark

StarCraft II Micromanagement (SMAC) is a popular benchmark in MARL [4, 5, 7, 40, 39]. We conduct experiments in 17 benchmark tasks of StarCraft II, which contains 14 popular tasks proposed by SMAC [8] and three more super hard cooperative tasks proposed by QPLEX [7]. In the micromanagement scenarios, each unit is controlled by an independent agent that must act based on its own local observation, and the enemy units are controlled by a built-in AI.

For evaluation, we compare EMC with the state-of-the-art algorithms: RODE [40], QPLEX [7], MAVEN [15], and the two variants of QMIX [5]: CW-QMIX and OW-QMIX [39]. All experimental results are illustrated with the median performance and 25-75% percentiles. Figure 6

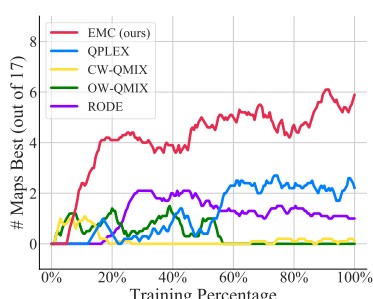

Figure 6: The number of scenarios in which the algorithm's median test win rate is the highest by as least 1/32.

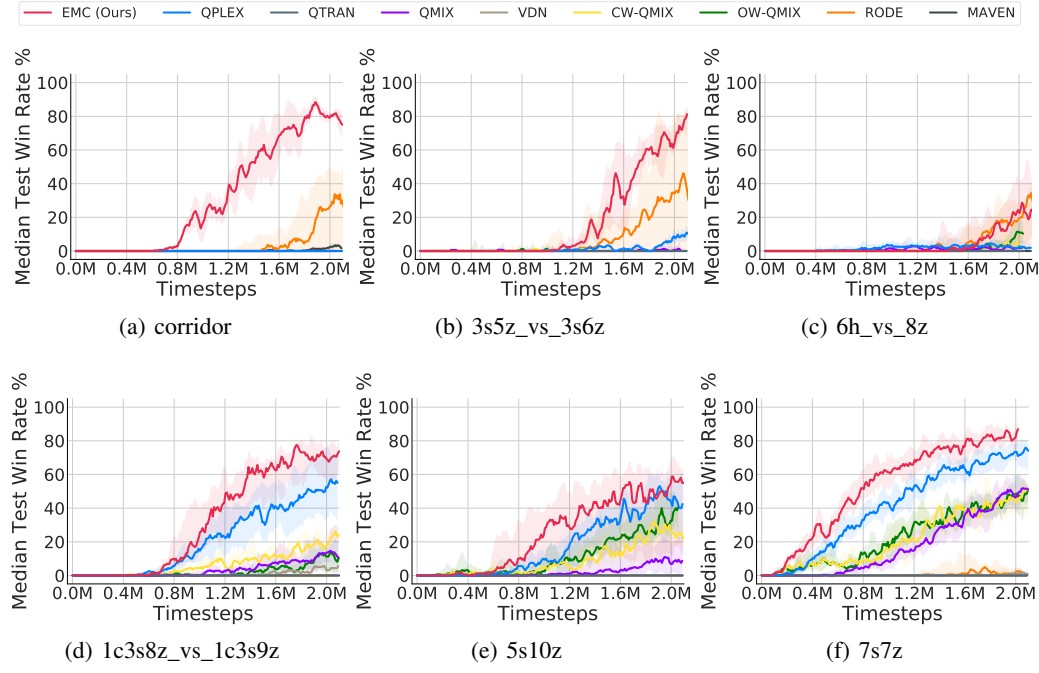

Figure 7: Results of super hard maps in SMAC.

shows the overall performance of the tested algorithms in all these 17 maps. Due to the effective exploration with episodic memory which can efficiently use promising exploratory experience trajectories, EMC is the best performer on up to 6 tasks, underperforms on just three tasks, and ties for the best performer on the rest tasks.

The advantages of our algorithm can be mainly illustrated by the results of the six hard maps which need sufficient exploration shown in Figure 7. The three maps in the first row are super hard, and solving them needs efficient, diverse and coordinated exploration. Thus, we can find that the EMC algorithm significantly outperforms other algorithms in *corridor* and *3s5z_vs_3s6z*, and also achieves the best performance (equal to RODE) in *6h_vs_8z*. To the best of our knowledge, this will be the state-of-the-art results in *corridor* and *3s5z_vs_3s6z*. For the remaining three maps in the second row ( *1c3s8z_vs_1c3s9z*, *5s10z*, and *7s7z*), where other baselines can also find winning strategies, due to the boost learning process via episodic memory along with efficient exploration, our algorithm EMC still performs the best in the three maps, with fastest learning speed and the highest rates achieved.

## 5.4 Ablation Study

To understand the superior performance of EMC, we carry out ablation studies to test the contribution of its two main components: curiosity module and episodic memory. Following methods are included in the evaluation: (i) EMC without curiosity module (denoted by *EMC-wo-C*); (ii) EMC without episodic memory component (denoted by *EMC-wo-M*); (iii) QPLEX, which can be considered as EMC without the episodic memory component nor the curiosity module, provides a natural ablation baseline of EMC.

Figure 8(b-c) shows that in easy exploration maps, both EMC and EMC-wo-C achieve the state-of-the-art performance, which implies that in the easy tasks, sufficient exploration can be achieved simply by the popular $\epsilon$-greedy method. However, in super hard exploration maps (Figure 8 (a)), EMC-wo-C cannot solve this task but EMC has excellent performance. These empirical experiments show that the curiosity module plays a vital role in improving performance when sufficient and coordinated exploration is necessary. On the other hand, making the best use of good trajectories collected by exploration is also essential. As shown Figure 8, EMC with episodic memory enjoys better sample efficiency than EMC-wo-M in challenging (Figure 8a) and easy exploration tasks (Figure 8(b-c)). In general, the curiosity module and the episodic memory complement each other, and efficiently using promising exploratory experience trajectories leads to the superior performance of EMC.

Like single-agent curiosity or RND [12] exploration methods, our approach looks simple yet effective. In addition, its design choices do not look straightforward before we know how to do it right.

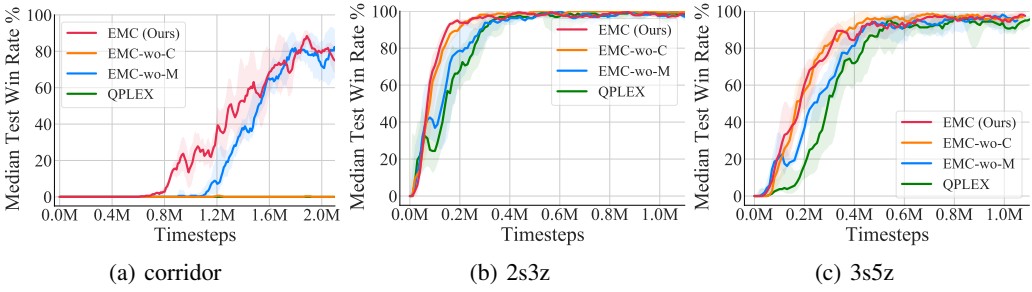

(a) corridor      (b) 2s3z      (c) 3s5z

Figure 8: Ablation study on the two major components.

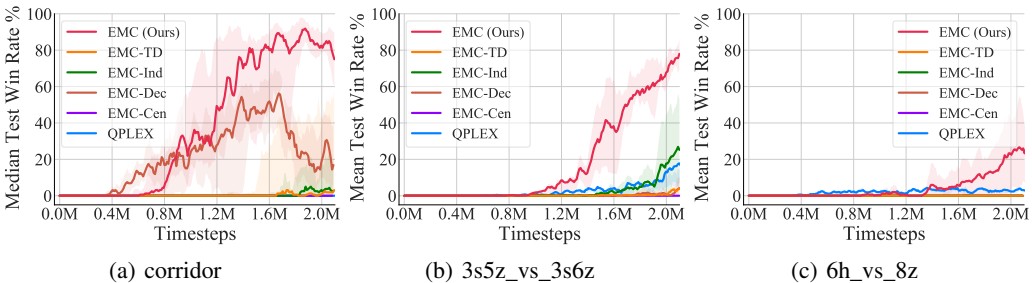

(a) corridor      (b) 3s5z_vs_3s6z      (c) 6h_vs_8z

Figure 9: Ablation study on design choice.

Therefore we conduct additional ablation studies to demonstrate the effect of our elaborate formulation of curiosity bias. We introduce several baselines and compare them with EMC : (i) using the normalized TD-error of $Q_{total}$ as curiosity rewards, denoted as *EMC-TD*; (ii) using the averaged error between the individual utilities and their targets as intrinsic rewards, denoted as *EMC-Ind*; (iii) using the TD error of a centralized critic of the controllers which conditions on all agents' histories and actions, denoted as *EMC-Cen*; (iv) using the averaged prediction errors of $Q_i^{ext;dec}$ which are trained in a decentralized way, denoted as *EMC-Dec*. We aim to investigate the subtle implementation difference between EMC and EMC-TD as well as EMC-Ind, and compare the exploration efficiency of our method with the global curiosity-driven exploration method (EMC-Cen) and local curiosity-driven exploration method (EMC-Dec) empirically.

We design a variant of the toygame mentioned in section 5.1, which has an additional random noisy reward region. By visualizations, we demonstrate that the agents of EMC-TD and EMC-Ind tend to get stuck in the noisy-reward region, thus resulting in sub-optimal policy, while our method show superior ability for avoiding such noise-spike problem. On the other hand, since EMC-Cen is based on global curiosity, which encourages agents to explore the whole state space without bias, it may fail in finding sparse but valuable interaction patterns in the exponentially growing space in complex tasks. When comparing with EMC and EMC-Dec, we find that the key difference is the counterfactual baseline (Eq. (2)), which can theoretically reduce the variance of EMC [41]. Therefore, EMC can focus more on the individual specific contribution and achieve the significant improvements.

We test these baselines in SMAC and the results are shown in Figure 9, and our method significantly outperform other baselines. In general, by conducting these ablations, we demonstrate the robustness for noise spikes of our design choice ((i) and (ii)), as well as the efficiency and stability of our method compared with centralized or decentralized curiosity-driven exploration method. More detailed discussions will be deferred to Appendix E.

## 6 Conclusions and Future Work

This paper introduces EMC, a novel episodic multi-agent reinforcement learning algorithm with a curiosity-driven exploration framework that allows for efficient coordinated exploration and boosted policy training by exploiting explored informative experiences. Based on the effective exploration ability, our method shows significant outperformance over state-of-the-art MARL baselines on challenging tasks in the StarCraft II micromanagement benchmark. The limitation of our work lies in the lack of adaptive exploration methods to ensure robustness. Besides, the episodic memory may get problems in stochastic settings. For future work, we may conduct further research in these directions.

## Acknowledgments and Disclosure of Funding

We would like to thank the anonymous reviewers for their valuable comments and helpful suggestions. This work is supported in part by Science and Technology Innovation 2030 – "New Generation Artificial Intelligence" Major Project (No. 2018AAA0100900), a grant from the Institute of Guo Qiang, Tsinghua University, and a grant from Turing AI Institute of Nanjing.

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
