# A Experiment Settings and Implementation Details

## A.1 StarCraft II

The benchmark we considered in our paper is the popular combat scenario of StarCraft II unit micromanagement tasks [8]. In this game, the enemy units are controlled by the built-in AI, and each ally unit is controlled by the reinforcement learning agent. We use the default settings, and the results in our paper use Version SC2.4.6.2.69232. At each time-step, each agent will choose action from the discrete action space, which includes the following actions: no-op, move [direction], attack [enemy id], and stop. By taking actions, agents move and attack in continuous maps. During the game, all agents will receive a global reward equal to the total damage done to enemy units. The team will get additional bonuses of 10 by killing each enemy unit, and bonuses of 200 by winning the combat. Here we briefly introduce each map of the SMAC challenges in Table 1.

| Map Name | Ally Units | Enemy Units |
|---|---|---|
| 2s3z | 2 Stalkers & 3 Zealots | 2 Stalkers & 3 Zealots |
| 3s5z | 3 Stalkers & 5 Zealots | 3 Stalkers & 5 Zealots |
| 1c3s5z | 1 Colossus, 3 Stalkers & 5 Zealots | 1 Colossus, 3 Stalkers & 5 Zealots |
| 5m_vs_6m | 5 Marines | 6 Marines |
| 10m_vs_11m | 10 Marines | 11 Marines |
| 27m_vs_30m | 27 Marines | 30 Marines |
| 3s5z_vs_3s6z | 3 Stalkers & 5 Zealots | 3 Stalkers & 6 Zealots |
| MMM2 | 1 Medivac, 2 Marauders & 7 Marines | 1 Medivac, 2 Marauders & 8 Marines |
| 2s_vs_1sc | 2 Stalkers | 1 Spine Crawler |
| 3s_vs_5z | 3 Stalkers | 5 Zealots |
| 6h_vs_8z | 6 Hydralisks | 8 Zealots |
| bane_vs_bane | 20 Zerglings & 4 Banelings | 20 Zerglings & 4 Banelings |
| 2c_vs_64zg | 2 Colossi | 64 Zerglings |
| corridor | 6 Zealots | 24 Zerglings |
| 5s10z | 5 Stalkers & 10 Zealots | 5 Stalkers & 10 Zealots |
| 7sz | 7 Stalkers & 7 Zealots | 7 Stalkers & 7 Zealots |
| 1c3s8z_vs_1c3s9z | 1 Colossus, 3 Stalkers & 8 Zealots | 1 Colossus, 3 Stalkers & 9 Zealots |

Table 1: SMAC challenges.

## A.2 Didactic Examples

Figure 3 shows the referred didactic example in section 5.1, which is a $11 \times 12$ grid world game. The blue agent and red agent can choose one of the five actions: *[up, down, left, right, stay]* at each time step. The two agents is isolated by the wall, and they cannot be observed by the other one until they get into the $5 \times 6$ light shaded area. They will receive a global positive reward of 10 if and only if they arrive at the dark shaded grid at the same time. If only one arrives, the incoordination will be punished by a negative reward of $-p$.

## A.3 Implementation Details

We adopt the PyMARL [8] implementation of state-of-the-art baselines: RODE [40], QPLEX [7], MAVEN [15], Qtran [6],VDN [4],QMIX [5] and Weighted-QMIX [39]. The hyper-parameters of these algorithms are the same as that in SMAC [8] and referred in their source codes. Our method is also based on QPLEX, and the hyper-parameters are the same referred in its source codes. While the special hyper-parameters are illustrated in Table 2 and other common hyper-parameters are adopted by the default implementation of PyMARL [8].

We conduct experiments on an NVIDIA Tesla P100 GPU, and each task in SMAC needs to train about 20 hours to 30 hours, depending on the number of agents and episode length limit of each map. We evaluate 32 episodes with decentralized greedy action selection without $\epsilon - greedy$ strategy every 10k timesteps for each algorithm. The test win rate shows the percentage of episodes in which agents defeat all enemy units within the time limit. Besides, since the intrinsic rewards need to vanish as the policy converges, we use a decaying weighting term to scale the intrinsic rewards:

| EMC's architecuture configurations | Value |
|---|---|
| soft update weight | 0.05 |
| weighting term $\lambda$ of episodic loss | 0.1 or 0.01 |
| episodic memory capacity | 1M |
| episodic latent dim | 4 |

Table 2: The hyper-parameters of EMC's architecture.

$\tilde{r}_t^{int} = \eta_t * r_t^{int}, \eta_{t+200k} = 0.9 * \eta_t$. In the three super hard maps: corridor, 3s5z_vs_3s6z, 6h_vs_8z, we set $\eta = 0.05$ while set $\eta = 0.0001$ in other maps.

## B  Experiments on StarCraftII

Figure 10 shows the performance of 6 easy maps in SMAC. It can be found that our algorithm perform the best in 5 of the 6 easy maps. In the map *2s_vs_1sc*, although EMC is the second best, the performance gap between EMC and the QPLEX algorithm is very subtle. The advantage of EMC over the other algorithms can be found in Figures 10(b), 10(c), and 10(f), where it converges much faster than the second best algorithm QPLEX. For example, in the *bane_vs_bane* tasks, the EMC algorithm reaches a $100\%$ win rate in fewer than $0.2M$ steps, while the QPLEX algorithm converges at the time step of $0.3M$. Futhermore, the win rate of QPLEX does not reach $100\%$ in this map and its learning process is not as stable as that of EMC. Therefore, although the state-of-the-art algorithms such as QPLEX performs sufficiently well in these easy maps, the coordinated exploration mechanism and episodic-memory control equipped by EMC can further enhance the performance a learning algorithm.

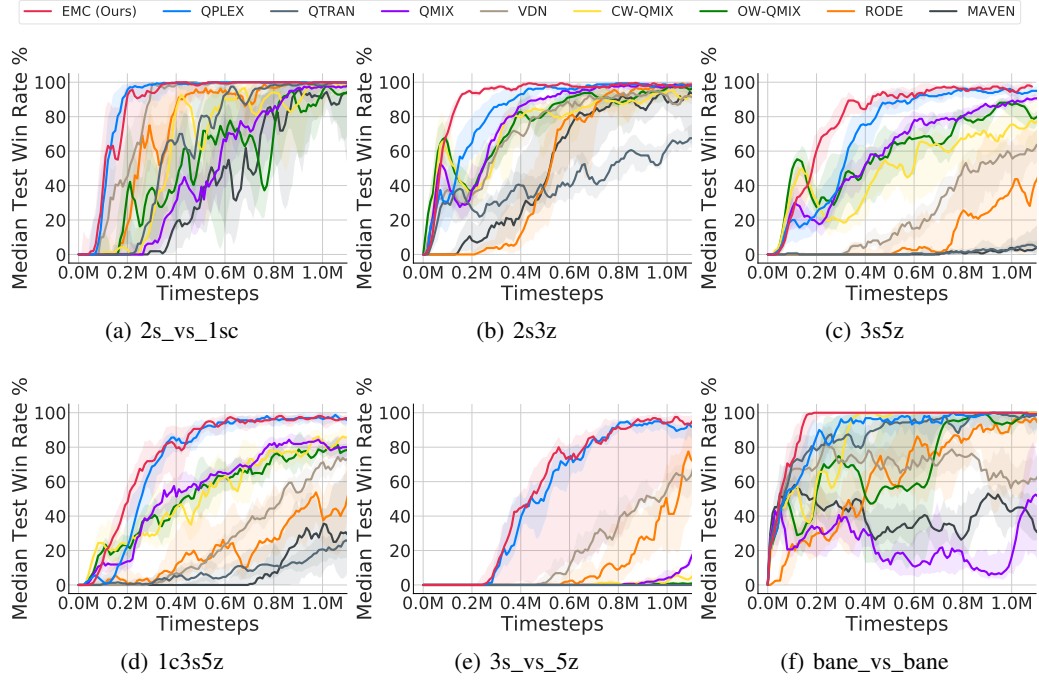

Figure 10: Results of the 6 easy maps in the SMAC experiments.

The results in Figure 11 shows that the EMC algorithm achieves comparble performance with the state-of-the-art baseline algorithms in the 5 maps *2c_vs_64zg*, *27m_vs_30m*, *MMM2*, *5m_vs_6m*, and *10m_vs_11m*. Moreover, our method may not perform pretty well in few maps, and we hypothesis that the training process of EMC may get stuck in local optimal due to the episodic control mechanism. It also should be noted that the MAVEN algorithm, which is specially designed for the multi-agent

exploration problem, performs the worst in the these maps. And our algorithm, which is also equipped with an exploration mechanism, outperforms MAVEN drastically.

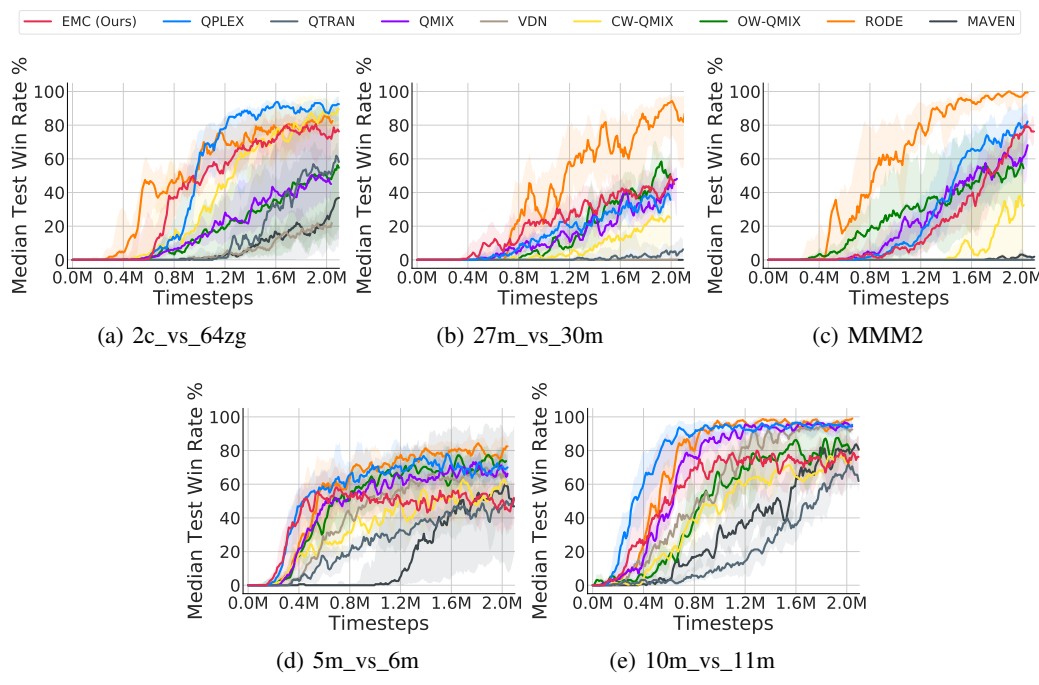

Figure 11: Results of the remaining 5 maps in the SMAC experiments .

# C   Experiments on the Coordinated Toygame

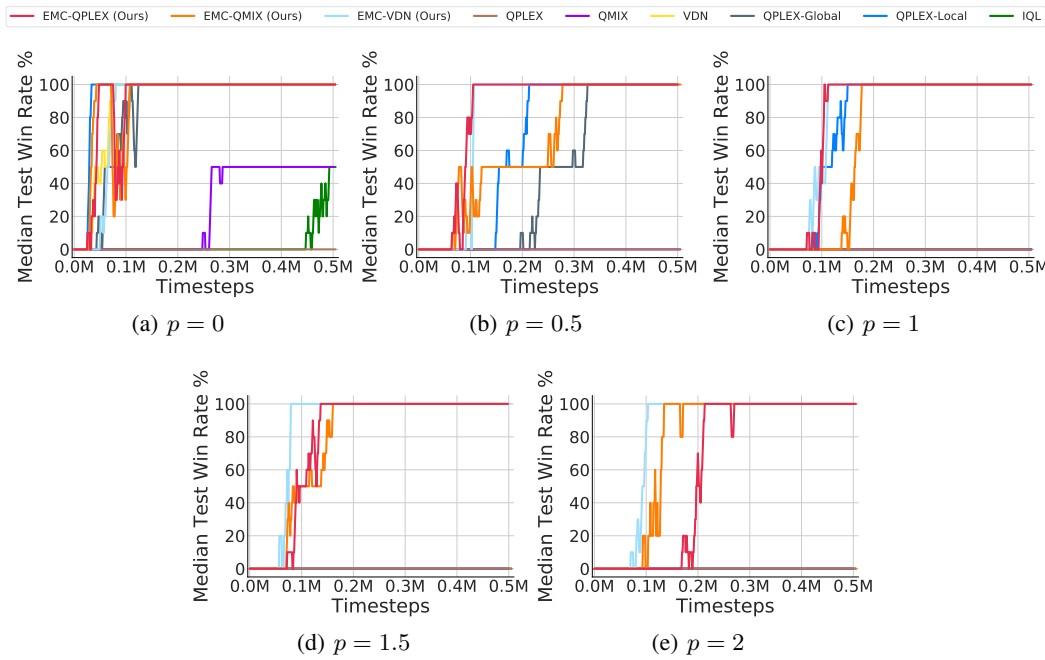

Figure 12: Results of the coordinated toygame with different punishment $p$ .

To compare EMC's ability of coordinated exploration with other algorithms, we conduct several experiments with different punishment degrees, i.e., different $p$, in the coordinated toygame (see Section 5.1). Figure 12 shows, when $p = 0$, all algorithms, except for QPLEX and IQL, can find winning strategies quickly since the game is easy. However, under the incoordination penalty ($p > 0$), algorithms without intrinsic motivation fail to win the game (Figure 12 (b)), since sufficient coordinated exploration needs to be addressed. Moreover, as expected, QPLEX-Local has an advantage over sample efficiency compared with QPLEX-Global (Figure 12 (b-c)) because of the decentralized exploration, which can avoid searching the whole state space. As $p$ increases (Figure 12 (d-e)), thanks to the biased and efficient exploration by predicting individual Q-values, only our methods can solve the task. By these experiments, we can conclude that neither centralized (global) curiosity nor decentralized (local) curiosity is practical for exploration in MARL. In contrast, predicting individual Q-values can capture the sparse and valuable interactions by leading agents to explore the areas where Q-values are more dynamic, thus achieve coordinated exploration effectively.

## D   Ablation Study of Coefficient Term

In this section, we study the different coefficient term $\lambda$ in Eq. (8) to visualize the robustness of our hyperparameters. The weighting term $\lambda$ of memory TD loss (see Section 5) was selected in {0.001, 0.01, 0.1, 0.5}. We study the influence of different $\lambda$ in several maps, i.e., *2s3z*, *3s5z*, and *3s5z_vs_3s6z*. Figure 13 shows that EMC with $\lambda = 0.01$ or $0.1$ can achieve the state-of-the-art performance. From these empirical experiments, we find that in general, $\lambda$ is not sensitive for most maps when chosen from $0.01 \sim 0.1$. However, the performance may be degenerated if $\lambda$ is too large (e.g., $\lambda = 0.5$ in *3s_vs_5z*) since the best memorized return of our episodic memory may bring learning into local optimum.

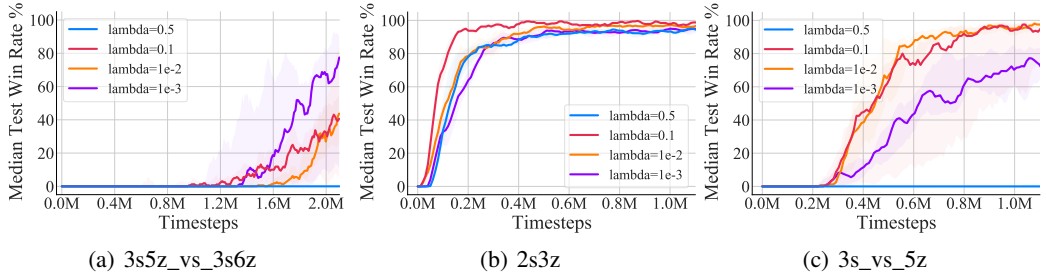

Figure 13: Ablation study of coefficient term $\lambda$.

## E   Ablation Study of Design Choice

The proposed intrinsic reward is the average of individual MSE (Eq. (3)) between utilities "Targets" and "Predictors", thus seems to be similar with the average of the Target-utilities individual error (which is just the normalized VDN TD-error). So we take a closer look and investigate the subtle difference between these two implementations. We first discuss the similarities between prediction error (EMC) and TD error, then we analyze their fundamental (high-level) difference and technical difference, and, finally, we provide empirical results to support our claims.

**Similarities**

There are some similar properties between prediction error and TD error: 1) they both converge when the policy converges; 2) they are common metrics that show promising results for exploration [12, 11, 42, 16] and exploitation [43–45], respectively. These similarities motivate us to study the effects of EMC's each module by comparing with the ablation study using TD error.

**Fundamental High-level Difference**

A fundamental (high-level) difference is that using TD error as an intrinsic reward can hurt performance since the objective of this intrinsic reward (maximize the TD error) is totally the opposite of the original objective of VDN (minimize the TD error). Intuitively, the metric of TD error does

not match the TD-learning framework in the perspective of objective functions. In contrast, we use prediction errors of individual Q-values (i.e., the embeddings of local action-observation histories) as intrinsic rewards for coordinated exploration. To stabilize training, we utilize a slowly updated target network for individual Q-values as a moving target. Using prediction error as a curiosity for exploration has been a long-standing topic in the single-agent deep RL literature [12, 11, 42, 16], and one important related work is [16], which measures the prediction error in the learnable representation space (i.e., this latent space is also a moving target). The literature [12, 11, 42, 16] implies that the idea of using prediction error is empirically effective, but the novelty of our EMC method is to situate this idea in multi-agent reinforcement learning by exploiting its factorization structure.

**Technical Difference**

More concretely, we will clarify the technical difference between our method (EMC) and the baseline using TD error as intrinsic rewards, as the reviewer suggested. As stated in Eq. (3) in Section 4, EMC (our method) uses the prediction error of the individual utilities as below:

$$r_{EMC}^{int} = \frac{1}{N} \sum_{i=1}^{N} \left\| \widetilde{Q}_i(\tau_i, \cdot) - Q_i^{target}(\tau_i, \cdot) \right\|_2.$$ (9)

While the normalized VDN TD-error (denoted as *EMC-TD*) can be formulated by:

$$
\begin{aligned}
r_{TD}^{int} &= \frac{1}{N} \left\| Q^{tot}(\boldsymbol{\tau}, \boldsymbol{a}) - \left( r^{ext} + \gamma \max_{\boldsymbol{a}'} Q_{tot}^{target}(\boldsymbol{\tau}', \boldsymbol{a}') \right) \right\|_2 \\
&= \frac{1}{N} \left\| \sum_{i=1}^{N} \left( Q_i(\tau_i, \cdot) - \left( r^{ext} + \gamma \max_{a_i'} Q_i^{target}(\tau_i', a_i') \right) \right) \right\|_2
\end{aligned}
$$ (10)

where $\boldsymbol{\tau}'$ denotes the joint history on the next state. If we ignore the difference in the summation operator, the major difference in Eq. (9) and Eq. (10) is two-fold:

a. TD error uses a one-step temporal difference which is involved with the immediate reward $r^{ext}$.

b. TD error uses the VDN utility functions $Q_i(\tau_i, \cdot)$, while EMC uses the predictors $\widetilde{Q}_i(\tau_i, \cdot)$.

Therefore, using TD errors as intrinsic rewards may result in the following several issues:

- Since it uses the one-step TD target with immediate reward $r^{ext}$, it can be sensitive to noise spikes (e.g. when rewards are stochastic), which can be exacerbated by bootstrapping, where approximation errors appear as another source of noise [44].

- Instead of predictors $\widetilde{Q}_i(\tau_i, \cdot)$ which are optimized end to end with the targets, VDN utility functions $Q_i$ are learned by one-step reward backpropagation, resulting in that the errors shrink slowly and the agents tend to be stuck in early trajectories. As discussed similarly in [44], the lack of diversity will make the system prone to over-fitting.

**Ablation Study**

To investigate the difference between EMC and EMC-TD, we carry out an ablation study on SMAC and a gridworld game. To study the major difference in Eq. (9) and Eq. (10) discussed above (i.e., (a) and (b) bullets), we introduce another baseline using $r_{ind}^{int}$ (denoted as *EMC-Ind*), which uses the averaged error between the individual utilities and their targets as intrinsic rewards.

$$r_{Ind}^{int} = \frac{1}{N} \sum_{i=1}^{N} \left\| Q_i(\tau_i, \cdot) - Q_i^{target}(\tau_i, \cdot) \right\|_2.$$ (11)

EMC-Ind aligns the individual utilities $Q_i(\tau_i, \cdot)$ and their target $Q_i^{target}(\tau_i, a_i)$ in the same temporal steps and does not include the predictor network $\widetilde{Q}_i(\tau_i, \cdot)$. Comparing EMC-Ind with EMC-TD and EMC, we can provide the ablation studies for the effect of (a) and (b), respectively. In the gridworld game, we combine the different curiosity methods (i.e., EMC, EMC-TD, and EMC-Ind) with the VDN learning algorithm and conduct the baseline VDN to demonstrate the effect of these intrinsic rewards in exploration. The experiments are listed as follows.

To better demonstrate the issues caused by (a) and (b), we introduce a new variant of the original gridworld task in the paper (Figure 14), which adds a noisy reward region above the shaded area, and two agents will receive a random Gaussian ($\mu = 0$, $\sigma = \{0, 0.0001, 0.01, 0.25\}$) noisy reward here. This noisy reward area is used to represent noise spikes in the reward function. The two agents need to jump out the local optimal area (the noisy reward region) and arrive at the goal grid at the same time.

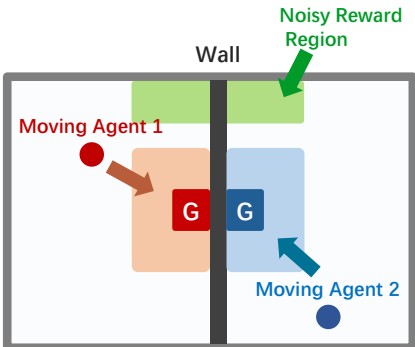

Figure 14: New grid world.

The results in Figure 15(a-d) show that EMC-VDN has achieved the best performance and EMC-Ind performs better than EMC-TD. The comparison between EMC-Ind and EMC-TD indicates that one step TD target is sensitive to noise spikes. The reason why EMC-Ind underperforms EMC-VDN is that EMC uses predictors $\tilde{Q}_i$ which are optimized end to end with the targets $Q_i^{target}$, while EMC-Ind uses utility functions $Q_i$ which are optimized with the one-step TD target. Therefore, the intrinsic reward of EMC-VDN will decay as the frequency of visiting the state-action pairs increases (i.e., capturing the novelty of states). In contrast, for EMC-Ind, the optimization of $Q_i$ is influenced by the one-step TD target which is softly updated by a fixed rate, thus the intrinsic rewards cannot vanish along with the number of training steps on the corresponding states (i.e., cannot capture the novelty of states). In other words, the prediction errors of EMC-Ind (i.e., the intrinsic rewards) depend on the update frequency rather than the novelty of visited states. Thus as the scale of noise increases, EMC-Ind and EMC-TD both fail in finding the optimal policy, and EMC-VDN significantly outperforms these two baselines. VDN cannot solve these problems, which indicates that the intrinsic reward introduced by EMC-VDN, EMC-TD, and EMC-Ind is effective for hard exploration problems.

For more clear clarifications, we provide the proportion of the visitation in the areas of noisy-reward and goal grid in the gridworld with $\sigma = 0.25$, respectively, and the results are shown in Figure 16. As we expected, the results show that, due to reasons discussed above, the agents of EMC-Ind or EMC-TD tend to get stuck in this noisy-reward region, and EMC-VDN can jump out of the local optimal area and reach the goal grid. Compared with EMC-Ind, the ability to capture state novelty provides EMC-VDN with more efficient curiosity-driven exploration. On the other hand, especially compared with VDN, EMC-TD will be stuck in the noisy-reward region longer, which shows that TD-error cannot perform well under higher noise spikes.

Figure 17 show the results of SMAC. We can see that EMC-Ind shows advantages over EMC-TD in corridor and 3s5z_vs_3s6z map, demonstrating that using one-step difference may harm performance. However, the winning rate of EMC-Ind shows relatively low compared with EMC (ours), indicating that it may be stuck in local optima.

As discussed above, we choose to use prediction errors as intrinsic rewards instead of using TD error, which can capture the dynamics of $Q_i$ quickly, avoid the noise spikes problem, and jump out of the local optima effectively. Empirical results show that EMC can encourage the agents to visit novel and promising states efficiently.

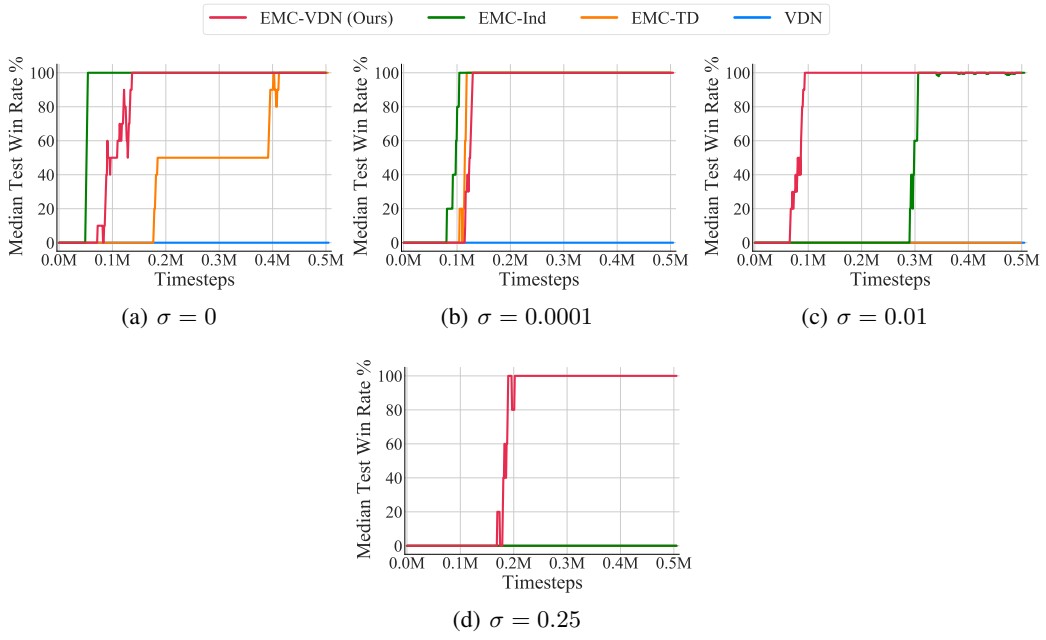

Figure 15: Results of the coordinated toygame with different scale of noisy reward .

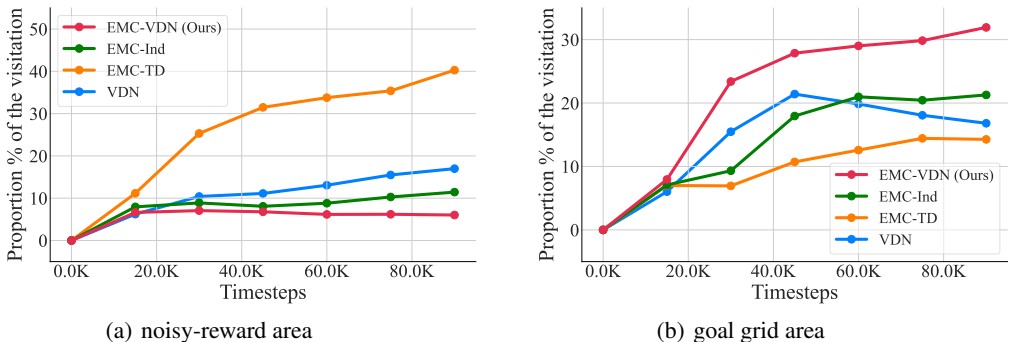

Figure 16: Proportion % of the visitation in the noisy-reward area and around the goal grid area ($\sigma = 0.25$), respectively.

## F    Ablation Study of Episodic Memory

To illustrate the ability of EMC in the stochastic setting, we conduct an ablation study by introducing stochasticity into the gridworld didactic task. In the original gridworld, each agent can move in four directions or stay still at each time step. In the stochastic variant, each agent has a $\xi\%$ probability of making a mistake and choose a random action accordingly. Figure 18(a-e) shows the performance of EMC and other baselines under different degrees of stochasticity ($\xi = \{0, 5, 10, 20, 30, 50\}$). These empirical results show that with proper stochasticity (e.g., $\xi = \{0, 5, 10, 20\}$), EMC can also significantly outperform baselines and solve this hard exploration puzzle. When the environment has a lot of uncertainty (e.g., $\xi = 30, 50$), it is challenging for EMC and other baselines. Learning in the highly stochastic environment is also an interesting future direction for MARL.

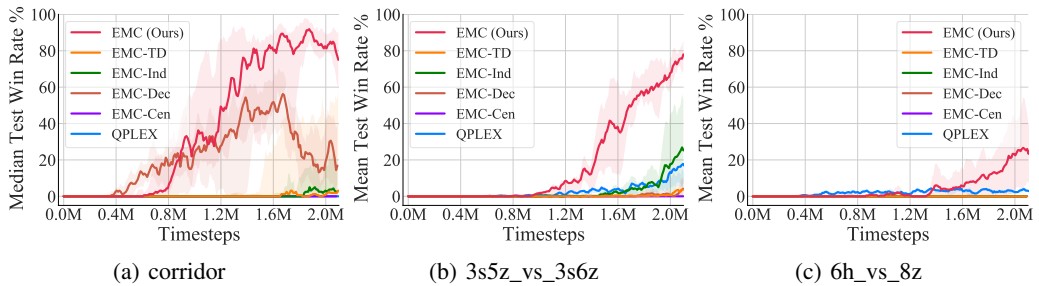

Figure 17: Ablation study on design choice.

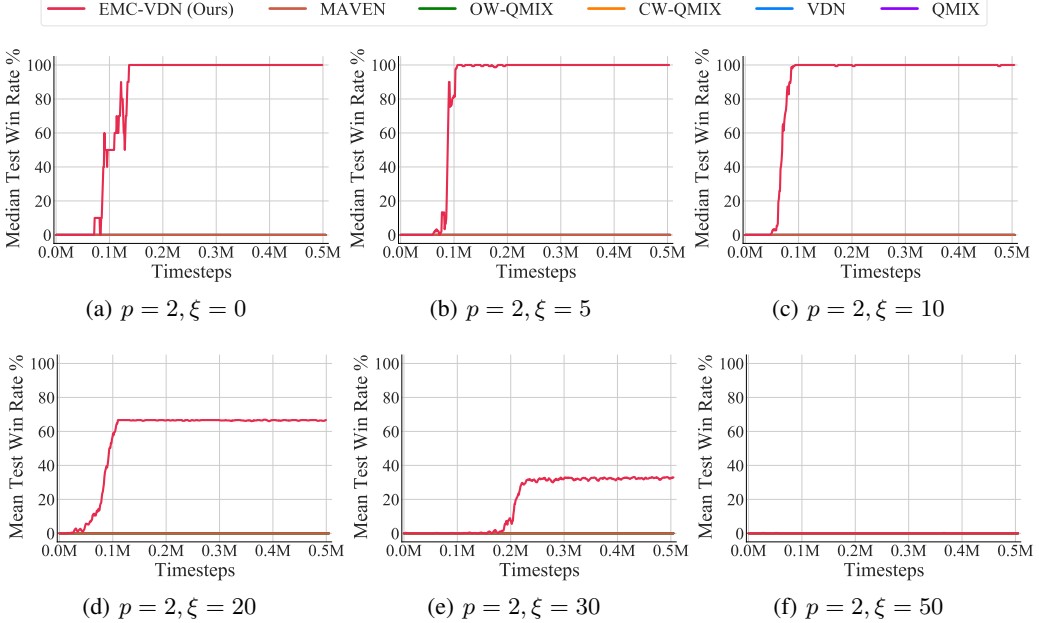

Figure 18: Results of the coordinated toygame with different scale of noisy reward .