# OpenReview forum: "Episodic Multi-agent Reinforcement Learning with Curiosity-driven Exploration"
_NeurIPS.cc/2021/Conference — NeurIPS 2021 Poster_

### Official Review · Reviewer_XAps · 2021-07-12

**Rating:** 7
**Confidence:** 4

**Summary:**

This work focuses on efficient exploration for MARL in the Centralized Training Decentralized Execution (CTDE) framework. It proposes to rely on two mechanisms to improve it: an intrinsic motivation based on the prediction error of the agents' Q-values and the use of episodic memory to store the highest return obtained from previously visited states.

**Limitations And Societal Impact:**

Quite rapidly, it could be developed.

**Main Review:**

2. Strong points
* The problem tackled by this work is important and well-motivated.
* The proposed method performs well and can be applied to any MARL algorithm with Q-value factorization.
* The method is tested against several baselines on several environments of varying difficulties. Visualizations are provided in a simple environment.
* Ablation studies are presented.

3. Possible improvements
      1. The motivation behind this choice of intrinsic motivation could be made clearer. From what I understand it uses Random Network Distillation (RND) but with a changing target, the individual Q-values, that are learned by TD-learning. Therefore it encourages visiting less-visited states (at least in the beginning) but more importantly states for which the Q values have changed (change in the strategy, environment, etc.). It seems that this would not be specific to multi-agent learning or value factorization but could also be applied to single-agent RL. It would therefore be interesting to motivate why the MARL setting, in particular, is considered. Also, it would be interesting to assess how much the centralized training of the Q-values impacts the performance of the exploration. In addition to QPLEX-Global and QPLEX-Local an additional baseline could be to train the $Q^{ext}_i$ in a decentralized way and use the sum of their prediction error as intrinsic motivation. This should test how much the "counterfactual baseline" term of Eq (2) plays a role in the efficiency of the curiosity.
    2. The implementation of the Episodic Memory with the memory table and the state projection could be made clearer. Also, could you explain why using the max return over previous trajectories does not break the learning? Indeed using the return of a previous outdated trajectory estimates the value under an outdated policy/team strategy. Could it be because it uses the max and that the optimal policy/team strategy should have the highest value for any state?
    3.   How many additional parameters and updates are required to use the Curiosity module (training all the additional Q-network and predictors)?
    4.   This work mentions "coordinated" exploration several times. Could you define what is meant, how to measure it, and how the proposed method improves it?


4. Recommendation
I think the paper is good but can be improved by addressing some points raised in 3.

5. Additional Remarks
    1. Could you comment on the use of an intrinsic reward based on the prediction error between Q-values trained in a centralized /factored way from TD and Q-values trained in a decentralized way (also from TD)
    2. The notation seems to alternate between $Q_{tot}^{all}$ and $Q_{tot}$ which is confusing. Also, the $Q$ values sometimes depend on the state and sometimes on the trajectory history.

**Time Spent Reviewing:**

4,5 hours

---

> ### Author Response · Authors · 2021-08-10
> **Response for reviewer XAps**
>
>  Thanks for your thoughtful comments. We provide clarification to your questions and concerns as below. We appreciate if you have any further questions or comments.
>
> **Q1:** It would therefore be interesting to motivate why the MARL setting.
>
> **A1:** Different from the single-agent setting, MARL faces the challenges of scalability and partial observability. As the number of agents increases, the state space is exponentially growing, and original exploration methods may fail. Therefore, our motivation is to provide a scalable and effective multi-agents exploration approach to deal with these problems, specifically in MARL. The curiosity module in our work takes advantage of a popular value factorization structure in the centralized training with decentralized execution (CTDE) paradigm of MARL, which shows both simplicity and effectiveness in multi-agents exploration problems. In addition, to our best knowledge, this paper is the first work that designs effective curiosity-driven exploration for cooperative MARL and achieves state-of-the-art performance on the SMAC benchmark.
>
>
> **Q2:**  The additional baseline that train the $Q_i^{ext}$ in a decentralized way.
>
> **A2:** Thanks for your insightful advice, and we add an additional baseline that trains the $Q_i^{ext}$ in a decentralized way (denoted by $Q_i^{ext;dec}$) and use the sum of their prediction errors as intrinsic motivation, denoted as *EMC-Decentralized*. Table (1-3) show that EMC-Decentralized can improve exploration ability in some maps (e.g., corridor), and EMC can still significantly outperform EMC-Decentralized. Compared with EMC and EMC-Decentralized, we find that the difference between $Q_i^{ext}$ and $Q_i^{ext;dec}$ is a counterfactual baseline (Eq. 2), which can theoretically reduce the variance of $Q_i^{ext}$ for EMC [1]. Therefore, EMC can more focus on the individual specific contribution (see Section 4.1) and reflect the novelty of states. Empirical results show that EMC provides more effective and stable exploration ability than EMC-Decentralized.
>
>
>
>
> Table 1: The median test win rate % in corridor map.
>
> |  Algorithms\Steps (Milion)   | 0M | 0.4M | 0.8M|1.2M|1.4M|1.6M|1.8M|2.0M|
> |  ----  | ----  |----  | ----  |----  | ----  |----  | ----  |----  |
> | **EMC (Ours)**| 0.0 |0.0| 2.1|31.4|52.2|66.9|77.3|81.9|
> |**EMC-Decentralized**| 0.0| 0.0|15.6|21.8|53.1|37.5|15.6|21.8|
> |**QPLEX**|0.0|0.0| 0.0 |0.0|0.0|0.0|0.0|0.0|
>
> Table 2: The median test win rate % in 3s5z_vs_3s6z map.
>
> |  Algorithms\Steps (Milion)   | 0M | 0.4M | 0.8M|1.2M|1.4M|1.6M|1.8M|2.0M|
> |  ----  | ----  |----  | ----  |----  | ----  |----  | ----  |----  |
> | **EMC (Ours)**| 0.0 |0.0| 0.0 |3.1|22.9|30.5|60.4|81.6|
> |**EMC-Decentralized**| 0.0| 0.0|0.0|0.0|0.0|0.0|0.05|0.09|
> |**QPLEX**|0.0 |0.0| 0.0|0.0|0.1|0.4|0.2|5.2|
>
> Table 3: The median test win rate % in 6h_vs_8z map.
>
> |  Algorithms\Steps (Milion)   | 0M | 0.4M | 0.8M|1.2M|1.4M|1.6M|1.8M|2.0M|
> |  ----  | ----  |----  | ----  |----  | ----  |----  | ----  |----  |
> | **EMC (Ours)**| 0.0 |0.0| 0.0|0.0|1.2|2.3|19.3|28.9|
> |**EMC-Decentralized**| 0.0| 0.0|0.0|0.0|0.0|0.0|0.0|0.0|
> |**QPLEX**|0.0|0.1| 0.1 |0.2|0.1|0.4|0.5|0.6|
>
>
>
> **Q3:** Why using the max return over previous trajectories does not break the learning?
>
> **A3:** First of all, in the deterministic environments, since the optimal policy has the highest value for any state, the max return used by episodic memory provides an effective lower bound and speed up the learning while still providing theoretical optimality guarantee in tabular cases [2]. Second, in the near-deterministic environments, episodic memory can provide effective learning with bounded error [2], and it empirically works well in many previous works [3,4,5,6].
>
>
> **Q4:** How many additional parameters and updates are required to use the curiosity module?
>
> **A4:** The parameters of the curiosity module are comparable with the learning algorithm in the inference module, and both of them depend on the number of agents. The updates frequency is also the same as the inference module.
>
>
> **Q5:** How to measure coordinated exploration and how the proposed method improves it?
>
> **A5:** Coordinated exploration means the agents can coordinate to jump out of the local optimal and explore to get a higher return. It is challenging to measure the pattern of coordinated exploration explicitly. Instead, if the agents can leverage the interactions from other agents and the environment, they can coordinate and achieve efficient exploration. In Section 5.1, we use a didactic example to illustrate our coordinated exploration. The heatmaps of intrinsic rewards show that predicting individual Q-values will bias exploration into areas where individual Q-values are more dynamic due to the potential correlation between agents. Besides, the median test win rate shows that our curiosity module can encourage agents to jump out of the local optima. Therefore, we claim that EMC can help improve coordinated exploration.
>
>
> [1] Wang, Y., Han, B., Wang, T., Dong, H., and Zhang, C. Off- policy multi-agent decomposed policy gradients. ICLR 2021.
>
> [2] Hu H, Ye J, Ren Z, et al. Generalizable Episodic Memory for Deep Reinforcement Learning[C]. ICML 2021.
>
> [3] Charles Blundell, Benigno Uria, Alexander Pritzel, Yazhe Li, Avraham Ruderman, Joel Z Leibo, Jack Rae, Daan Wierstra, and Demis Hassabis. Model-free episodic control. arXiv preprint arXiv:1606.04460, 2016.
>
> [4] Steven S Hansen, Pablo Sprechmann, Alexander Pritzel, André Barreto, and Charles Blundell. Fast deep reinforcement learning using online adjustments from the past. In Proceedings of the 32nd International Conference on Neural Information Processing Systems, pages 10590–10600, 2018.
>
> [5] M Lengyel and P Dayan. Hippocampal contributions to control: The third way. In Twenty-First Annual Conference on Neural Information Processing Systems (NIPS 2007), pages 889–896. Curran, 2008.
>
> [6] Alexander Pritzel, Benigno Uria, Sriram Srinivasan, Adria Puigdomenech Badia, Oriol Vinyals, Demis Hassabis, Daan Wierstra, and Charles Blundell. Neural episodic control. In International Conference on Machine Learning, pages 2827–2836. PMLR, 2017.

---

> > ### Comment · Reviewer_XAps · 2021-08-30
> > **thank you for the authors' response**
> >
> > I think that the authors addressed well my concerns so I have decided to raise my score.

---

> > > ### Author Response · Authors · 2021-09-01
> > > **Thanks for your response**
> > >
> > > Thank you very much for the insightful and positive comments!  We  appreciate the reviewer’s interest and efforts in the deep understanding of our work, which is scientific and also helpful to further improve our paper. We will incorporate relevant discussions and experiments of the rebuttal into the next revision.Thanks again for your time and efforts!

---

### Official Review · Reviewer_vJkd · 2021-07-16

**Rating:** 6
**Confidence:** 3

**Summary:**

The paper presents a solution for improving exploration in multi-agent environments. Episodic Multi-agent reinforcement learning with Curiosity-driven exploration (EMC) adds a curiosity and episodic memory module on top of standard value-factorized Q-learning algorithms typically used in multi-agent control (e.g. QMix). First, an intrinsic reward is defined through the curiosity module. The curiosity module trains separate Q-functions from the policy to predict the Q-values only on external reward. In addition to these external-reward Q-functions, the curiosity module also has a collection of "target" Q-functions. The intrinsic curiosity reward is then derived as the scaled value of the L2 distance between target and external-reward Q-functions. The motivation behind this particular intrinsic reward is that each individual agent's external-reward Q-values contain information from every other agent due to the mixing function used to estimate the final combined Q-value, meaning that coordinated exploration is more likely to occur with this intrinsic reward than others based only on local state information. Additionally to the curiosity module, EMC uses an episodic memory which, similar to work on episodic control, stores the maximum obtained Monte-Carlo return of a particular state. Using a non-parametric look-up into the episodic memory provides the learning process the ability to regularize its target Q-function to an observed Monte-Carlo return.

Results are demonstrated on a toy environment, Predator-Prey and SMAC, where EMC outperforms other baseline methods especially in environments where exploration is a significant factor towards success. An analysis on the toy environment additionally shows in more detail the improved coordinated exploration afforded by EMC.

**Limitations And Societal Impact:**

The paper describes some limitations, but as mentioned in the main review, the episodic memory's sensitivity to outliers is one limitation that I foresee could have impact in non-stationary and highly stochastic environments.

I don't believe there is large potential for negative societal impact, as the method is only a curiosity-based intrinsic reward for multi-agent reinforcement learning.

**Main Review:**

The paper's proposed method, EMC, provides a straightforward and effective solution to the problem of curiosity-based exploration / intrinsic motivation for multi-agent reinforcement learning. The curiosity module resembles Random Network Distillation (RND) and other related methods, where the error between target and predictor network output represents the intrinsic reward provided to the agent. In EMC, the use of an external-reward Q-network as the target for the prediction is an interesting contribution by itself. It could seemingly also be applied to the setting of single-agent environments, where incorrect predictions of the Q-value by the predictor networks could signal the agent towards re-visiting states where its Q-values are changing more rapidly, which seems like a useful signal for learning. The paper convincingly motivates their formulation of the intrinsic reward both by (1) demonstrating that the per-agent Q-functions capture global information via the use of a mixing function during Q-learning and (2) extensive comparisons against a variety of baselines on both toy and standard benchmarks.

The other contribution of EMC, which is the episodic memory component, is maybe less well-motivated. Episodic control methods are potentially brittle to highly stochastic environments, since in those settings a stored maximum return is likely to be an outlier, i.e. caused by lucky environment dynamics and not necessarily due to an agent's own actions. Episodic control's sensitivity to outliers could be especially noticeable in multi-agent environments, which have multiple learning systems interacting in non-stationary ways. The ablations in Figure 8 reveal a maybe minor effect of the episodic memory component on learning speed, and an insignificant effect on final performance (although the figures are only on 3 of 17 SMAC tasks).

It would make the paper stronger if the authors can (1) expand more on the motivation of the memory component, (2) demonstrate that this memory is useful even with stochastic environments (or otherwise show that this isn't a serious problem more generally) and (3) have clearer empirical results demonstrating the performance improvement of the memory component (e.g. maybe ablations over the components, presented over all scenarios similar to Figure 6). Even with these unanswered questions about the episodic memory, I still rate the paper as accept due to the curiosity module representing an effective solution to multi-agent coordinated exploration.

Also, is there a reason that MAVEN, another exploration method for multi-agent RL mentioned in the paper, is missing from certain figures (Figures 4, 5, 6).

In terms of writing, the paper is clear and well-written. The method's explanation is easily understandable, with Figure 2 providing a direct overview. The connections to previous work are sufficiently highlighted.

**Time Spent Reviewing:**

4

---

> ### Author Response · Authors · 2021-08-10
> **Response for reviewer vJkd (part I)**
>
> Thanks for your insightful comments. We provide clarification to your questions and concerns as below. If you have any further questions or comments, please post them and we will be happy to have further discussions.
>
> **Q1:** Expand more on the motivation of the memory component.
>
> **A1:** The episodic memory can make the best use of good trajectories collected by the curiosity module and improve the sample efficiency. Concretely, in the early stage of training, our method can rapidly latch onto past successful strategies by the non-parametric episodic memory and thus provide a reference point for boosting learning. Using the maximum return from the episodic memory to propagate rewards, we can compensate for the disadvantage of slow-learning resulted from the original one-step reward update and thus improve sample efficiency.
>
> **Q2:** The performance and analyse of the episodic memory in a stochastic setting?
>
> **A2:** If the environment is deterministic, it is theoretically convergent and optimal in tabular cases [1]. On the other hand, in the near-deterministic environment, the episodic memory can provide effective learning with bounded error [1], and it empirically works well in many previous works [2,3,4,5]. In the end, episodic memory may fail in a totally stochastic environment. For better illustration, we carry out an ablation study on the stochastic variant of the gridworld task. In the original gridworld, each agent can move in four directions or stay still at each time step. In the stochastic variant, each agent has a $\xi$\% probability of making a mistake and choose a random action accordingly. Table (1-4) show the performance of EMC and other baselines under different degrees of stochasticity $(\xi=\{0, 5, 10, 50\})$. These empirical results show that with proper stochasticity (e.g., $\xi=\{0, 5, 10\}$), EMC can also significantly outperform baselines and solve this hard exploration puzzle. When the environment has a lot of uncertainty (e.g., $\xi=50$), it is challenging for EMC and other baselines. Learning in a highly stochastic environment is also an interesting future direction for MARL.
>
> Table 1: The median test win rate % in gridworld ($p=2,\xi=0$).
>
> |  Algorithms \Steps (Milion) | 0M | 0.1M | 0.2M|0.3M|0.4M|0.5M|
> |  ----  | ----  |----  | ----  |----  | ----  |----  |
> | **EMC-VDN (Ours)**| 0.0 |100.0| 100.0|100.0|100.0|100.0|
> |**MAVEN**| 0.0| 0.0|0.0|0.0|0.0|0.0|
> |**OW-QMIX** | 0.0| 0.0|0.0|0.0|0.0|0.0|
> |**CW-QMIX**| 0.0| 0.0|0.0|0.0|0.0|0.0|
> |**VDN**| 0.0| 0.0|0.0|0.0|0.0|0.0|
> |**QMIX**| 0.0| 0.0|0.0|0.0|0.0|0.0|
>
>
>
> Table 2: The median test win rate % in gridworld ($p=2,\xi=5$).
>
> |  Algorithms \Steps (Milion) | 0M | 0.1M | 0.2M|0.3M|0.4M|0.5M|
> |  ----  | ----  |----  | ----  |----  | ----  |----  |
> | **EMC-VDN (Ours)**| 0.0 |0.0| 100.0|100.0|100.0|100.0|
> |**MAVEN**| 0.0| 0.0|0.0|0.0|0.0|0.0|
> |**OW-QMIX** | 0.0| 0.0|0.0|0.0|0.0|0.0|
> |**CW-QMIX**| 0.0| 0.0|0.0|0.0|0.0|0.0|
> |**VDN**| 0.0| 0.0|0.0|0.0|0.0|0.0|
> |**QMIX**| 0.0| 0.0|0.0|0.0|0.0|0.0|
>
> Table 3: The median test win rate % in gridworld ($p=2,\xi=10$).
>
> |  Algorithms \Steps (Milion) | 0M | 0.1M | 0.2M|0.3M|0.4M|0.5M|
> |  ----  | ----  |----  | ----  |----  | ----  |----  |
> | **EMC-VDN (Ours)**| 0.0 | 0.0|0.0|0.0|0.0|100.0|
> |**MAVEN**| 0.0| 0.0|0.0|0.0|0.0|0.0|
> |**OW-QMIX** | 0.0| 0.0|0.0|0.0|0.0|0.0|
> |**CW-QMIX**| 0.0| 0.0|0.0|0.0|0.0|0.0|
> |**VDN**| 0.0| 0.0|0.0|0.0|0.0|0.0|
> |**QMIX**| 0.0| 0.0|0.0|0.0|0.0|0.0|
>
> Table 4: The median test win rate % in gridworld ($p=2,\xi=50$).
>
> |  Algorithms \Steps (Milion) | 0M | 0.1M | 0.2M|0.3M|0.4M|0.5M|
> |  ----  | ----  |----  | ----  |----  | ----  |----  |
> | **EMC-VDN (Ours)**| 0.0 | 0.0|0.0|0.0|0.0|0.0|
> |**MAVEN**| 0.0| 0.0|0.0|0.0|0.0|0.0|
> |**OW-QMIX** | 0.0| 0.0|0.0|0.0|0.0|0.0|
> |**CW-QMIX**| 0.0| 0.0|0.0|0.0|0.0|0.0|
> |**VDN**| 0.0| 0.0|0.0|0.0|0.0|0.0|
> |**QMIX**| 0.0| 0.0|0.0|0.0|0.0|0.0|
>
> **Q3:** Clearer empirical results demonstrating the performance improvement of the memory component
>
> **A3:** We provide the ablation study of the total 17 maps of SMAC. The tables below show the median test rate of EMC, the median test rate of EMC-wo-M, and the median test rate percentage increase during the training process, respectively. For example, in map 2s3z at 0.2M steps, the median test rate of EMC is 87.5%, while that of EMC-wo-M is 75.0%, then the percentage increase is $\frac{87.5-75}{75}=16.67\\%$. The results show that EMC significantly outperforms EMC-wo-M. The superior performance of EMC implies that episodic memory plays a vital role in improving sample efficiency, especially in the early training stage.
>
> Table 5: The median test win rate % in 17 maps.
>
> |  Maps\Steps (Milion)    | 0M | 0.2M | 0.4M|0.6M|0.8M|1.0M|
> |  ----  | ----  |----  | ----  |----  | ----  |----  |
> |**2s_vs_1sc**|0, 0, 0%| 96.8,100,  -3.13%| 100,100,  0%|100, 100, 0%|100 ,100 ,0%|100,100,0%|
> | **2s3z** |0, 0, 0%| 87.5,75.0, 16.67%| 98.43, 96.87,  1.61%|95.31,100,  -4.69%| 98.44,100, -1.56%| 96.88 ,96.88, 0%|
> |**3s5z**|0, 0, 0%| 62.5, 31.25,  100%|0.875, 71.00, 21.73%|0.84,0.90,  -6.89%|0.97,0.97,0%|0.96, 0.9,  6.9%|
> |**1c3s5z**|0, 0, 0%|46.9, 25.0,  87.5%| 71.9, 78.1,  -8.0%|96.9,90.6, 6.9%|96.9, 96.9, 0.0%|96.9,93.7, 3.3%|
> |**3s_vs_5z**|0, 0, 0%|0, 0, 0%| 37.5,50.0,  -25%|78.1,7.12, 0%|84.3,96.8, -12.0% |93.7,90.6, 3.4%||
> |**bane_vs_bane**|0, 0, 0%|100.0, 87.5, 14%| 100,100, 0% |100, 100, 0%|100,100, 0%|100,100, 0%|
>
>
> |  Maps\Steps (Milion)    | 0M | 0.4M | 0.8M|1.2M|1.6M|2.0M|
> |  ----  | ----  |----  | ----  |----  | ----  |----  |
> |**1c3s8z_vs_1c3s9z**|0, 0, 0%|0, 0, 0%|6.250,  7.81,   -20% |50.0, 17.18, 190.90%|65.62, 42.18, 55.55%|71.87, 53.12,  35.29%|
> |**5s10z**|0, 0, 0%|0, 0, 0%| 3.12, 6.25, -50% |28.12, 25.00,  12.5%|50.00, 50.00,  0.0%|56.25, 50.00, 12.5%|
> |**7s7z**|0, 0, 0%|28.125, 12.5, 125%| 43.75, 32.81, 33.33% |71.87, 53.12,  35.29%|81.25,  54.68,  48.0%|87.50, 70.31, 24.4%|
> |**3s5z_vs_3s6z**|0, 0, 0%|0, 0, 0%| 0, 0, 0% |3.1,0, \\| 30.50, 20.50, 48.78%|81.60,64.10, 27.30%|
> |**6h_vs_8z**|0, 0, 0%|0, 0, 0%| 0, 0, 0%|0, 0, 0%|2.30, 0.30,  666.67%|28.9,  21.10, 36.97%|
> |**corridor**|0, 0, 0%|0, 0, 0%| 2.1, 0,  \\|31.40, 9.50,  230.52%|66.90,  63.20,  5.85% |81.90, 73.60,  11.27%|
>
> |  Maps\Steps (Milion)    | 0M | 0.4M | 0.8M|1.2M|1.6M|2.0M|
> |  ----  | ----  |----  | ----  |----  | ----  |----  |
> |**2c_vs_64zg**|0, 0, 0%|0,  0,  0%| 50.0,15.62,  220.0%|56.25, 84.37, -33.33%| 71.87, 90.63, -20%|62.50, 90.625, -31.03%|
> |**27m_vs_30m**|0, 0, 0%|3.12, 0, \\| 15.62, 6.25,  150% |40.625, 12.5, 225%|34.37, 28.12,  22.22%|45.612, 38.9, 17.2%|
> |**MMM2**|0, 0, 0%|0, 0, 0%| 3.125, 1.56,  100% |15.625, 13.43,  16.34%|50.0,65.62,  -23.80|78.12, 85.93,  -9.09%|
> |**5m_vs_6m**|0, 0, 0%|50.0, 40.62,  23.07%| 46.875, 62.5,  -25.0% |37.5, 71.87, -47.82%|50.0, 65.62, -23.80%|46.87, 71.87, -34.78%|
> |**10m_vs_11m**|0, 0, 0%|15.62, 70.31, -77.77%| 71.87, 90.62, -20% |78.12, 92.18,  -15.25%|75.0, 96.87, -22.58%|75.00, 93.75, -20.0%|
>
> \\: when the median test win rate of EMC-wo-M is zero, the percentage increase has no meaning.
>
> **Q4:** The performance of MAVEN in gridworld task and the predator-prey task.
>
> **A4:** We add the evaluation of MAVEN in the gridworld and predator-prey tasks, where Table (5-7) illustrate the performance of MAVEN as well as WQMIX in the gridworld (miscoordination penalty $p=\{0,0.5,2\}$) task and the predator-prey task. Empirical results show the median test win rate of all methods over eight random seeds. Compared with MAVEN and WQMIX, EMC shows better exploration ability and significantly outperform MAVEN and WQMIX.
>
> Table 6: The median test win rate % in gridworld game (p=0).
>
> |  Algorithms\Steps (Milion)  | 0M | 0.1M | 0.2M|0.3M|0.4M|0.5M|
> |  ----  | ----  |----  | ----  |----  | ----  |----  |
> | **EMC-QPLEX (Ours)**| 0.0 |100.0| 100.0|100.0|100.0|100.0|
> | **CW-QMIX** | 0.0| 100.0|100.0|100.0|100.0|100.0|
> |**OW-QMIX**| 0.0| 50.0|100.0|100.0|100.0|100.0|
> |**MAVEN**| 0.0| 22.0|50.0|50.0|100.0|100.0|
> | **EMC-wo-C** | 0.0| 0.0|0.0|0.0|0.0|0.0|
> |**EMC-wo-M**| 0.0| 100.0|100.0|100.0|100.0|100.0|
> |**QPLEX**|0.0|0.0| 0.0 |0.0|0.0|0.0|
>
> Table 7: The median test win rate % in gridworld game (p=0.5).
>
> |  Algorithms\Steps (Milion)    | 0M | 0.1M | 0.2M|0.3M|0.4M|0.5M|
> |  ----  | ----  |----  | ----  |----  | ----  |----  |
> | **EMC-QPLEX (Ours)**| 0.0 |50.0| 100.0|100.0|100.0|100.0|
> | **CW-QMIX** | 0.0| 0.0|0.0|0.0|0.0|0.0|0.0|0.0|
> |**OW-QMIX**| 0.0| 0.0|0.0|0.0|0.0|0.0|0.0|0.0|
> |**MAVEN**| 0.0| 0.0|0.0|0.0|0.0|0.0|0.0|0.0|
> | **EMC-wo-C** | 0.0| 0.0|0.0|0.0|0.0|0.0|0.0|0.0|
> |**EMC-wo-M**| 0.0| 50.0|85.0|100.0|100.0|100.0|
> |**QPLEX**|0.0|0.0| 0.0 |0.0|0.0|0.0|0.0|0.0|
>
> Table 8: The median test win rate % in gridworld game (p=2).
>
> | Algorithms \ Steps (Milion)   | 0M | 0.1M | 0.2M|0.3M|0.4M|0.5M|
> |  ----  | ----  |----  | ----  |----  | ----  |----  |
> | **EMC-QPLEX (Ours)**| 0.0 |50.0| 100.0|100.0|100.0|100.0|
> | **CW-QMIX** | 0.0| 0.0|0.0|0.0|0.0|0.0|0.0|0.0|
> |**OW-QMIX**| 0.0| 0.0|0.0|0.0|0.0|0.0|0.0|0.0|
> |**MAVEN**| 0.0| 0.0|0.0|0.0|0.0|0.0|0.0|0.0|
> | **EMC-wo-C** | 0.0| 0.0|0.0|0.0|0.0|10.0|
> |**EMC-wo-M**| 0.0| 50.0|80.0|90.0|90.0|100.0|
> |**QPLEX**|0.0|0.0| 0.0 |0.0|0.0|0.0|0.0|0.0|
>
> Table 9: The median test win rate % in predator-prey task.
>
> |  Algorithms\Steps (Milion)   | 0M | 0.2M | 0.4M|0.6M|0.8M|1.0M|
> |  ----  | ----  |----  | ----  |----  | ----  |----  |
> | **EMC-QPLEX (Ours)**| 0.0 |22.8| 25.7|24.8|37.8|40.0|
> |**MAVEN**| 0.0| 0.0|0.0|0.0|0.0|-0.875|
> | **EMC-wo-C** | 0.0| 0.0|0.0|0.0|0.0|0.0|0.0|0.0|
> |**EMC-wo-M**| 0.0| 0.0|12.4|15.6|21.7|30.8|
> |**QPLEX**|0.0|0.0| 0.0 |0.0|0.0|0.0|0.0|0.0|

---

> > ### Author Response · Authors · 2021-08-10
> > **Response for reviewer vJkd (part II)**
> >
> > [1] Hu H, Ye J, Ren Z, et al. Generalizable Episodic Memory for Deep Reinforcement Learning[C]. ICML, 2021.
> >
> > [2] Charles Blundell, Benigno Uria, Alexander Pritzel, Yazhe Li, Avraham Ruderman, Joel Z Leibo, Jack Rae, Daan Wierstra, and Demis Hassabis. Model-free episodic control. arXiv preprint arXiv:1606.04460, 2016.
> >
> > [3] Steven S Hansen, Pablo Sprechmann, Alexander Pritzel, André Barreto, and Charles Blundell. Fast deep reinforcement learning using online adjustments from the past. In Proceedings of the 32nd International Conference on Neural Information Processing Systems, pages 10590–10600, 2018.
> >
> > [4] M Lengyel and P Dayan. Hippocampal contributions to control: The third way. In Twenty-First Annual Conference on Neural Information Processing Systems (NIPS 2007), pages 889–896. Curran, 2008.
> >
> > [5] Alexander Pritzel, Benigno Uria, Sriram Srinivasan, Adria Puigdomenech Badia, Oriol Vinyals, Demis Hassabis, Daan Wierstra, and Charles Blundell. Neural episodic control. In International Conference on Machine Learning, pages 2827–2836. PMLR, 2017.

---

### Official Review · Reviewer_qZwR · 2021-07-16

**Rating:** 6
**Confidence:** 3

**Summary:**

This paper studies curiosity-driven exploration for multi-agent deep reinforcement learning. Specifically, instead of deploying centralized curiosity exploration or decentralized curiosity exploration,  the paper learns a factorized centralized critic which is a monotonic linear combination of individual agents’ Q value. Then, the prediction errors of individual agent’s Q-values are used as intrinsic rewards that encourage exploration. In addition, an episodic memory is used to prioritize promising experience trajectories and speedup training. The proposed approach is compared with competitive baselines on StarCraft II multi-agent challenges (SAMC).

**Ethical Concerns:**

The reviewer doesn't see any ethical issues with this paper.


**Limitations And Societal Impact:**

The authors discuss the limitations of their work. Discussion on societal impact seems missing.


**Main Review:**

The paper is well structured and easy to follow. The multi-agent exploration problem it aims to address is an important problem for our community. The experimental results look promising. However, the reviewer has some questions about the claims made in the papers. Please see the detailed comments below.


1. The authors claim that the proposed approach achieves better coordinated exploration. However, it is unclear why and how the proposed curiosity module improves coordination. Is it because the intrinsic reward is assigned to a global state and thus it encourages agents to transition to the global state? If it is the case, why the proposed approach could achieve better coordination than global curiosity exploration. Moreover, the author mentions that the proposed curiosity module could `biases exploration into states where strong interdependence may lie between agents’. It is unclear to the reviewer how the proposed approach could achieve this. Could you elaborate?


2. In MARL, the size of the global state space grows exponentially with the number of agents. Will the proposed curiosity module encourage agents to visit all states? If yes, could you comment on the scalability of the proposed approach?


3. This paper considers multi-agent exploration in a dense-reward setting. On the other hand, previous multi-agent exploration work such as EITI /  EDTI [1] considers sparse-reward setting, i.e. agents only receive rewards when completing a task, which is much more challenging than dense-reward setting. Experiments, analysis, or comments on the performance of the proposed approach on sparse-reward environments would make the submission stronger.


[1] Influence-Based Multi-Agent Exploration, Wang et. al. ICLR 2020

====== Post Rebuttal ======

Thanks for the reply.  The author response addressed my concern and the additional experimental results look good. I raise my score to 6.



**Time Spent Reviewing:**

5 hours

---

> ### Author Response · Authors · 2021-08-10
> **Response for  reviewer qZwR**
>
>  Thanks for your constructive comments. We provide clarification to your questions and concerns as below. If you have any further questions or comments, please post them and we will be happy to have further discussions.
>
> **Q1:**  Is it because the intrinsic reward is assigned to a global state and thus it encourages agents to transition to the global state?
>
> **A1:** No, the intrinsic reward does not encourage agents to transition to the global state. In fact, using global curiosity for exploration is ineffective since it encourages agents to visit all configurations without bias, leaving it difficult to find the critical locality interaction between agents, which seems too sparse compared with the exponentially growing state space when the number of agents increases. Please refer to Q2 for a detailed discussion.
>
> **Q2:** Why and how the proposed curiosity module improves coordination? Why can it achieve better coordination than global curiosity exploration?
>
> **A2:** Different from using global curiosity, individual Q-values $Q_i$ are the embeddings of historical observations, and are dynamically updated by the backpropagation of the global reward signal gained through cooperation during centralized training. Thus $Q_i$ can implicitly reflect the influence from the environment and other agents, and predicting $Q_i$ can capture valuable and spare interactions among agents. In a word, our method will focus on areas where the $Q_i$ are more dynamic instead of exploring the whole state space. Besides, Section 5.1 uses a didactic example to demonstrate this empirically. We also provide some additional results with baseline QPLEX-Global, EMC-wo-M, EMC-wo-C of some super hard maps in SMAC. The results are shown in Table 1. QPLEX-Global is based on global curiosity, which encourages agents to explore the whole state space, as mentioned in the paper. Table (1-3) show the empirical results that EMC significantly outperforms all baselines, illustrating that predicting dynamic $Q_i$ can guide agents to achieve more efficient coordinated exploration.
>
> Table 1: The median test win rate \% in corridor map.
>
> |  Algorithms\Steps (Milion) | 0M | 0.4M | 0.8M|1.2M|1.4M|1.6M|1.8M|2.0M|
> |  ----  | ----  |----  | ----  |----  | ----  |----  | ----  |----  |
> | **EMC (Ours)**| 0.0 |0.0| 2.1|31.4|52.2|66.9|77.3|81.9|
> |**QPLEX-Global**|0.0 |0.0| 0.0|0.0|0.0|0.0|0.0|0.0|
> | **EMC-wo-M** | 0.0 |0.0| 0.0|8.9|39.2|64.5|77.1|76.4|
> |**EMC-wo-C**|0.0 |0.0| 0.0|0.0|0.0|0.0|0.0|0.0|
> |**QPLEX**|0.0 |0.0| 0.0|0.0|0.1|0.4|0.2|5.2|
>
>
> Table 2: The median test win rate \% in 3s5z_vs_3s6z map.
>
> |  Algorithms\Steps (Milion) | 0M | 0.4M | 0.8M|1.2M|1.4M|1.6M|1.8M|2.0M|
> |  ----  | ----  |----  | ----  |----  | ----  |----  | ----  |----  |
> | **EMC (Ours)**| 0.0 |0.0| 0.0 |3.1|22.9|30.5|60.4|81.6|
> |**QPLEX-Global**|0.0 |0.0| 0.0|0.0|0.0|0.0|0.0|0.0|
> | **EMC-wo-M** | 0.0 |0.0| 0.0|0.0|10.2|20.5|45.4|64.1|
> |**EMC-wo-C**|0.0 |0.0| 0.0|0.0|0.1|0.4|0.3|4.1|
> |**QPLEX**|0.0 |0.0| 0.0|0.0|0.1|0.4|0.2|5.2|
>
> Table 3: The median test win rate % in 6h_vs_8z map.
>
> |  Algorithms\Steps (Milion)   | 0M | 0.4M | 0.8M|1.2M|1.4M|1.6M|1.8M|2.0M|
> |  ----  | ----  |----  | ----  |----  | ----  |----  | ----  |----  |
> | **EMC (Ours)**| 0.0 |0.0| 0.0|0.0|1.2|2.3|19.3|28.9|
> |**QPLEX-Global**|0.0 |0.0| 0.0|0.0|0.0|0.0|0.0|0.0|
> | **EMC-wo-M** | 0.0| 0.0|0.0|0.0|0.2|0.3|14.2|21.1|
> |**EMC-wo-C**| 0.0| 0.0|0.0|0.0|0.1|0.2|0.1|0.2|
> |**QPLEX**|0.0|0.1| 0.1 |0.2|0.1|0.4|0.5|0.6|
>
> **Q3:** Will the proposed curiosity module encourage agents to visit all states? and the scalability of the proposed approach?
>
> **A3:** No, the proposed curiosity module does not encourage agents to visit all states. In contrast with exploring all states, our method encourages agents to explore where the individual Q-values $Q_i$ are more dynamic, and try to leverage the potential interaction locality structure among agents which are reflected by the Q-values. Therefore, it is scalable and efficient since the Q-value space is much compact than the original state space.
>
> **Q4:** Experiments and analysis on the performance of EMC on a sparse reward setting?
>
> **A4:** In fact, our gridworld game is a sparse reward task. Since only when the two agents arrived at the goal grid at the same time, they can receive a global reward signal. In a sparse reward task that needs to explore a long sequence without extrinsic reward signals, at the beginning of exploration, the individual Q-values are not zeros there thanks to the random initialization of the neural networks. This leads to prediction errors which can be used as intrinsic rewards to encourage exploration. With more visitation in these places, the individual Q-values will be more accurate by training, resulting in the decrease of the intrinsic rewards. Therefore, without extrinsic rewards, the agents are also encouraged to visit places with high novelty (where Q-values are far from zeros), just like the trick of RND (Burda et al.). Due to the reason above, our method shows effective exploration ability in sparse reward settings like the gridworld task in Section 5.1.

---

> > ### Author Response · Authors · 2021-09-01
> > **Thanks for your feedback!**
> >
> > Thank you very much for the inspiring and insightful comments which really helped us to improve our work! We will incorporate the relevant discussions and experiments of the rebuttal in the next revision. Thanks again for your time and efforts!

---

### Official Review · Reviewer_rGKf · 2021-07-16

**Rating:** 5
**Confidence:** 4

**Summary:**

The paper proposes to use prediction errors of agent specific Q-functions as an intrinsic exploration bonus in multi-agent reinforcement learning (RL) similar to how prediction errors have been used in single agent RL. Another contribution is the application of episodic memory as an auxiliary loss in multi-agent RL. The proposed approach outperforms comparison methods in experiments.

**Limitations And Societal Impact:**

Yes

**Main Review:**

ORIGINALITY:

The proposed multi-agent RL approach is a novel combination of existing single agent techniques. To use curiosity and episodic memory in multi-agent RL the paper makes important design decisions and proposes a new framework.

The references require a thorough check. For example, references [18] and [19] refer to the same paper.


QUALITY:

The Q-functions $Q_{tot}^{ext}$, $Q_i^{ext}$, etc. used inside the curiosity module are implemented as neural networks separate from $Q_i$ and $Q_{tot}$. This design choice could be discussed in more detail. What would be potential problems if using the same Q-functions instead?

The paper should explain why in Equations (4) and (5) the Q-function has as parameter $s$. Under partial observability $s$ is not observed? On Line 209 $Q_{tot}$ has as parameter the history.

In the episodic memory approach, selecting the highest reward observed so far seems to be a very heuristic and application specific approach. Is there analysis (in the single agent case?) on when it does not work? For example, in stochastic environments very unlikely events but with high reward can mislead the exploration process into sub-optimal solutions. The intrinsic motivation based reward seems a less "dangerous" feature of the proposed approach since Q-function prediction errors should decrease with more samples. Appendix B shows that the episodic memory module which is a greedy heuristic leads easily to sub-optimal solutions in some benchmarks.

There are two components in the proposed approach, episodic memory and intrinsic reward. Why is the episodic memory used with the global state and with the global Q function but intrinsic reward is used with individual Q-functions dependent on action-observation histories? Should not similar reasoning apply to both?

What is the interplay of the episodic memory and intrinsic motivation? It was mentioned that intrinsic motivation is not sufficient on its own but the episodic memory is needed in addition. This connection could be explained in more detail.

In "As the dynamics of an agent’s individual Q-value function captures the novelty of states and the influence from other agents",
what does "dynamics of an agent’s individual Q-value function" mean exactly?

The proposed approach has several hyper-parameters due to two features: intrinsic reward and episodic memory. Please, specify how hyper-parameters were chosen and how much computational effort was used to find hyper-parameters?


CLARITY:

The paper is in many parts well written. Figure 2 is very useful.

The storyline in the intro is somewhat misleading. Centralized training with decentralized execution (CTDE) is not such a recent paradigm as advertised. Dec-POMDPs, where typical solutions involve computing a solution offline in a centralized manner and executing policies decentralized, have been around already for some time [Bernstein et al., 2002].

Bernstein, D. S., Givan, R., Immerman, N., & Zilberstein, S. (2002). The complexity of decentralized control of Markov decision processes. Mathematics of operations research, 27(4), 819-840.

On Line 121: "and attracted great attention.". What does this mean exactly and why is it important?

Line 194: What are "orediction errors"?

Line 219: Please, rephrase "how to use the best use of good trajectories collected by exploration".

Line 220: In "Recently, episodic control has become very popular", why is it importance that episodic control has become popular? Please, concrete statements: what are the benefits of episodic control and where do the benefits come from?

Line 222: "Inspires by this" should be "Inspired by this"?


SIGNIFICANCE:

Improving exploration in multi-agent RL is important. The idea of using ideas from single agent RL exploration based on curiosity and extending them to multi-agent RL individual agents is interesting. The way episodic memory is used to improve utilization of experience should be better motivated and the weaknesses discussed properly / replaced with something less heuristic and greedy. The proposed framework is partly well motivated but needs further clarifications or/and improvements in order for other researchers to be able to take advantage of it.

UPDATE:

The rebuttal addressed some of my concerns. I am still not completely satisfied with the theoretical motivation and will keep my score.


**Time Spent Reviewing:**

7

---

> ### Author Response · Authors · 2021-08-10
> **Response for Reviewer rGKf**
>
> Thanks for your constructive comments. We provide clarification to your questions and concerns as below. If you have any further questions or comments, please post them and we will be happy to have further discussions.
>
> **Q1:** The Q-functions $Q_{tot}^{ext}$, $Q_i^{ext}$, used inside the curiosity module are implemented as neural networks separate from $Q_i$ and $Q_{tot}$.What would be potential problems if using the same Q-functions?
>
> **A1:** The same Q-function may cause training instability. We have tested this implementation, i.e., using the same network to generate intrinsic rewards and inference. In fact, using the same Q-functions can also help coordinated exploration in the early stage of training. However, the prediction errors of the Q-values are used as intrinsic rewards acting on the same network, resulting in a loop in the training procedure. Thus the intrinsic rewards may increase sharply, resulting in divergent policy learning.
>
>
> **Q2:** Why does the Q-function have parameter $s$ in Equations (4) and (5)?
>
> **A2:** Because it is for the centralized training process. Our paper is based on the popular MARL paradigm, centralized training with decentralized execution (CTDE) [1], which is adopted to deal with scalability and partial observability. With this paradigm, agents' policies are trained with access to global information in a centralized way and executed only based on local histories in a decentralized way. In MARL, many algorithms adopt the paradigm of CTDE and allow the centralized training module has access to the global state $s$ (see QPLEX [2], QMIX [3], QTRAN [4]), while local agents cannot observe global states during the execution process.  Moreover, if the global state $s$ is inaccessible, MARL algorithms can use the joint histories of all local agents instead of $s$.
>
> **Q3:** The analysis of episodic memory on when it does not work.
>
> **A3:** First of all, when the environment is deterministic, it has theoretically optimal convergence in tabular settings [5]. On the other hand, in the near-deterministic environments, episodic memory can provide effective learning with bounded error [5], and empirically works well in many previous works [6,7,8,9]. In the end, episodic memory may fail in a totally stochastic environment. For better illustration, we carry out an ablation study on the stochastic variant of the gridworld task. In the original gridworld, each agent can move in four directions or stay still at each time step. In the stochastic variant, each agent has a $\xi$\% probability of making a mistake and choose a random action accordingly. Table (1-4) show the performance of EMC and other baselines under different degrees of stochasticity $(\xi=\\{0, 5, 10, 50\\})$. These empirical results show that with proper stochasticity (e.g., $\xi=\\{0, 5, 10\\}$), EMC can also significantly outperform baselines and solve this hard exploration puzzle. When the environment has a lot of uncertainty (e.g., $\xi=50$), it is challenging for EMC and other baselines. Learning in a highly stochastic environment is also an interesting future direction for MARL.
>
> Table 1: The median test win rate % in gridworld ($p=2,\xi=0$).
>
>  |  Algorithms \Steps (Milion) | 0M | 0.1M | 0.2M|0.3M|0.4M|0.5M|
> |  ----  | ----  |----  | ----  |----  | ----  |----  |
> | **EMC-VDN (Ours)**| 0.0 |100.0| 100.0|100.0|100.0|100.0|
> |**MAVEN**| 0.0| 0.0|0.0|0.0|0.0|0.0|
> |**OW-QMIX** | 0.0| 0.0|0.0|0.0|0.0|0.0|
> |**CW-QMIX**| 0.0| 0.0|0.0|0.0|0.0|0.0|
> |**VDN**| 0.0| 0.0|0.0|0.0|0.0|0.0|
> |**QMIX**| 0.0| 0.0|0.0|0.0|0.0|0.0|
>
>
>
>  Table 2: The median test win rate % in gridworld ($p=2,\xi=5$).
>
> |  Algorithms \Steps (Milion) | 0M | 0.1M | 0.2M|0.3M|0.4M|0.5M|
> |  ----  | ----  |----  | ----  |----  | ----  |----  |
> | **EMC-VDN (Ours)**| 0.0 |0.0| 100.0|100.0|100.0|100.0|
> |**MAVEN**| 0.0| 0.0|0.0|0.0|0.0|0.0|
> |**OW-QMIX** | 0.0| 0.0|0.0|0.0|0.0|0.0|
> |**CW-QMIX**| 0.0| 0.0|0.0|0.0|0.0|0.0|
> |**VDN**| 0.0| 0.0|0.0|0.0|0.0|0.0|
> |**QMIX**| 0.0| 0.0|0.0|0.0|0.0|0.0|
>
> Table 3: The median test win rate % in gridworld ($p=2,\xi=10$).
>
> |  Algorithms \Steps (Milion) | 0M | 0.1M | 0.2M|0.3M|0.4M|0.5M|
> |  ----  | ----  |----  | ----  |----  | ----  |----  |
> | **EMC-VDN (Ours)**| 0.0 | 0.0|0.0|0.0|0.0|100.0|
> |**MAVEN**| 0.0| 0.0|0.0|0.0|0.0|0.0|
> |**OW-QMIX** | 0.0| 0.0|0.0|0.0|0.0|0.0|
> |**CW-QMIX**| 0.0| 0.0|0.0|0.0|0.0|0.0|
> |**VDN**| 0.0| 0.0|0.0|0.0|0.0|0.0|
> |**QMIX**| 0.0| 0.0|0.0|0.0|0.0|0.0|
>
> Table 4: The median test win rate % in gridworld ($p=2,\xi=50$).
>
> |  Algorithms \Steps (Milion) | 0M | 0.1M | 0.2M|0.3M|0.4M|0.5M|
> |  ----  | ----  |----  | ----  |----  | ----  |----  |
> | **EMC-VDN (Ours)**| 0.0 | 0.0|0.0|0.0|0.0|0.0|
> |**MAVEN**| 0.0| 0.0|0.0|0.0|0.0|0.0|
> |**OW-QMIX** | 0.0| 0.0|0.0|0.0|0.0|0.0|
> |**CW-QMIX**| 0.0| 0.0|0.0|0.0|0.0|0.0|
> |**VDN**| 0.0| 0.0|0.0|0.0|0.0|0.0|
> |**QMIX**| 0.0| 0.0|0.0|0.0|0.0|0.0|
>
>
>
> **Q4:** Why is the episodic memory used with the global state and with the global Q function but intrinsic reward is used with individual Q-functions dependent on action-observation histories?
>
> **A4:** As we mentioned in Q3, since the episodic memory is part of the centralized training module, it can have access to the global state and utilize it for training. In MARL, the state space increases with the number of agents, resulting in exploration by global curiosity inefficient. Therefore, we leverage an insight of popular factorized MARL algorithms that the “induced" individual Q-values, which can capture localized interactions between agents. Based on the prediction errors of individual Q-values, we can derive the intrinsic reward inducing coordinated exploration to new or promising states.
>
> **Q5:** What is the interplay of the episodic memory and intrinsic motivation?
>
> **A5:** The episodic memory can make the best use of good trajectories collected by the intrinsic motivation module and improve the sample efficiency. Concretely, in the early stage of training, our method can rapidly latch onto past successful strategies by the non-parametric episodic memory and thus provide a reference point for boosting learning. Using the maximum return from the episodic memory to propagate rewards, we can compensate the disadvantage of slow-learning resulted by original one-step reward update and improve sample efficiency.
>
> **Q6:**  What does "dynamics of an agent’s individual Q-value function" mean exactly?
>
> **A6:** The dynamics of an agent’s individual Q-value function means the changing value of $Q_i$. The dynamics of $Q_i$ is caused by learning and interactions with the environment and other agents. Therefore, the dynamics of $Q_i$ can capture the novelty of states and the influence among agents.
>
> **Q7:** How hyper-parameters were chosen and how much computational effort was used to find hyper-parameters?
>
> **A7:**  We use grid search to choose hyper-parameters. The weight term of episodic loss was tuned comparing values of \{0.1, 0.01, 0.001\}, while the intrinsic rewards scale was tuned comparing values of \{0.2, 0.5, 1\}.
>
>
> [1] Bernstein, D. S., Givan, R., Immerman, N., \& Zilberstein, S. (2002). The complexity of decentralized control of Markov decision processes. Mathematics of operations research, 27(4), 819-840.
>
> [2] Jianhao Wang, Zhizhou Ren, Terry Liu, Yang Yu, and Chongjie Zhang. Qplex: Duplex dueling multi-agent q-learning. ICLR, 2021.
>
> [3] Tabish Rashid, Mikayel Samvelyan, Christian Schroeder, Gregory Farquhar, Jakob Foerster, and Shimon Whiteson. Qmix: Monotonic value function factorisation for deep multi-agent reinforcement learning. In International Conference on Machine Learning, pages 4295–4304, 2018.
>
> [4] Kyunghwan Son, Daewoo Kim, Wan Ju Kang, David Earl Hostallero, and Yung Yi. Qtran: Learning to factorize with transformation for cooperative multi-agent reinforcement learning. arXiv preprint arXiv:1905.05408, 2019.
>
> [5] Hu H, Ye J, Ren Z, et al. Generalizable Episodic Memory for Deep Reinforcement Learning[C]. ICML, 2021.
>
> [6] Charles Blundell, Benigno Uria, Alexander Pritzel, Yazhe Li, Avraham Ruderman, Joel Z Leibo, Jack Rae, Daan Wierstra, and Demis Hassabis. Model-free episodic control. arXiv preprint arXiv:1606.04460, 2016.
>
> [7] Steven S Hansen, Pablo Sprechmann, Alexander Pritzel, André Barreto, and Charles Blundell. Fast deep reinforcement learning using online adjustments from the past. In Proceedings of the 32nd International Conference on Neural Information Processing Systems, pages 10590–10600, 2018.
>
> [8] M Lengyel and P Dayan. Hippocampal contributions to control: The third way. In Twenty-First Annual Conference on Neural Information Processing Systems (NIPS 2007), pages 889–896. Curran, 2008.
>
> [9] Alexander Pritzel, Benigno Uria, Sriram Srinivasan, Adria Puigdomenech Badia, Oriol Vinyals, Demis Hassabis, Daan Wierstra, and Charles Blundell. Neural episodic control. In International Conference on Machine Learning, pages 2827–2836. PMLR, 2017.

---

> > ### Author Response · Authors · 2021-09-12
> > **Request Details**
> >
> > Thank you very much for your feedback! We are pleased that our response has addressed some of our concerns. However, we are not sure what theoretical motivation the reviewer is not satisfied with. We really appreciate it if the reviewer can provide more detailed information.
> >
> > Thanks again for your time and efforts in reviewing our work!
> >
> > The Authors

---

> > > ### Comment · Reviewer_rGKf · 2021-09-12
> > > **Episodic memory**
> > >
> > > The motivation for the episodic memory seems to be to limit the approach to specific kinds of problems. There are many ways for improving sample efficiency in specific tasks by adding limitations and greediness to MARL algorithms.
> > >
> > > Nevertheless, I agree that the authors have done a good job in the rebuttal and the paper has many other valuable parts. While my score is 5, that is, I want the episodic memory part to be removed (which would at the moment require a re-submission) since it does not add any value to the paper, I am not fighting for rejection.

---

> > > > ### Author Response · Authors · 2021-09-12
> > > > **Thanks for your feedback!**
> > > >
> > > > We would like to thank the reviewer for the thoughtful comments. The episodic memory plays an important role in EMC, which can effectively accelerate learning process by making the best use of the trajectories collected by the curiosity exploration, as discussed in the ablation study in Section 5.4 in detail. In fact, episodic memory is a general way for improving sample efficiency in single agent reinforcement learning, which shows promising results in previous works [1,2,3,4]. To the best of our knowledge, we are the first to utilize the mechanism of episodic control and show its capability in deep multi-agent setting, which may be inspiring for future works to improve sample efficiency for MARL.
> > > >
> > > > [1] Alexander Pritzel, Benigno Uria, Sriram Srinivasan, Adria Puigdomenech Badia, Oriol Vinyals, Demis Hassabis, Daan Wierstra, and Charles Blundell. Neural episodic control. In International Conference on Machine Learning, pages 2827–2836. PMLR, 2017.
> > > >
> > > > [2] Zichuan Lin, Tianqi Zhao, Guangwen Yang, and Lintao Zhang. Episodic memory deep q- networks. In IJCAI, 2018.
> > > >
> > > > [3] Zhu G, Lin Z, Yang G, et al. Episodic Reinforcement Learning with Associative Memory[C]//International Conference on Learning Representations. 2019.
> > > >
> > > > [4] Hu H, Ye J, Ren Z, et al. Generalizable Episodic Memory for Deep Reinforcement Learning[C]. ICML, 2021.

---

### Decision · Program_Chairs · 2021-09-28

**Decision:**

Accept (Poster)

**Comment:**

This paper proposes a new exploration algorithm specifically for multi-agent RL with factorized and centralized value function (critic). The idea is to use individual agent's value prediction errors as intrinsic rewards based on the intuition that this captures the influence from other agents thanks to the factorized/centralized aspect. In addition, the paper proposes an episodic memory that exploits past good experiences to further improve sample-efficiency.

Improving exploration in the context of multi-agent RL is also an important problem. All of the reviewers appreciated that the proposed idea is insightful and well-motivated, and the results are also strong. There is still a remaining concern regarding the effectiveness of the episodic memory especially in stochastic environments. The authors added a new result showing that the proposed method is still effective to some degree of stochasticity during the rebuttal period, and the reviewers are generally satisfied with this result. Therefore, I recommend accepting this paper and suggest the authors to include this for the camera-ready version.

**Consistency Experiment:**

NeurIPS has a long history of experimentation. In 2014, NeurIPS ran an experiment in which 10% of submissions were reviewed by two independent committees to quantify the randomness in the review process. This year, we repeated a variant of this experiment to see how the quality of the review process has changed over time.  This paper was part of the experiment and was therefore assigned to two committees (consisting of reviewers, an Area Chair, and a Senior Area Chair) that reached independent decisions.  If both committees made the same recommendation, this recommendation was followed. If a single committee recommended acceptance, the paper was accepted (with the exception of a few cases in which the other committee identified what we considered a fatal flaw, e.g., an error in a key result).

Both committees reached the same decision: **Accept (Poster)**

The other committee assigned to the paper recommended **Accept (Poster)**.  You can find the other set of reviews, along with any follow up discussion with the authors here:
https://openreview.net/forum?id=cLYyCXHU7g1n